# WholeBodyVLA: Towards Unified Latent VLA for Whole-Body Loco-Manipulation Control

**Haoran Jiang**[1,2,4,3*]  **Jin Chen**[1,2,4*]  **Qingwen Bu**[2]  **Li Chen**[2]  **Modi Shi**[4,2]
**Yanjie Zhang**[3]  **Delong Li**[3]  **Chuanzhe Suo**[3]  **Chuang Wang**[3]  **Zhihui Peng**[3†]  **Hongyang Li**[2†]

[1]Fudan University  [2]OpenDriveLab & MMLab at The University of Hong Kong  [3]AGIBOT  [4]SII
[*]Equal contribution  [†]Project co-lead

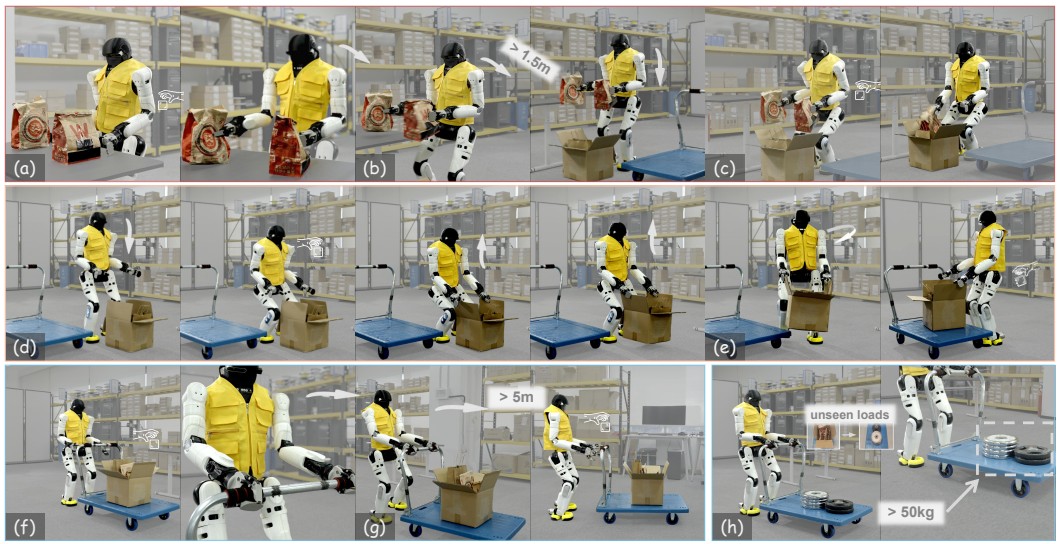

Figure 1: Introducing **WholeBodyVLA**, a humanoid system that operates on Agibot X2 robot and performs end-to-end humanoid loco–manipulation in large space for the first time. The proposed system achieves consecutive tasks autonomously, including **(a-c)** bimanual grasping, side-step towards the box, and squatting to place; **(d-e)** squatting to grasp and lift the box, and turning to place the box onto the cart; **(f-h)** grasping the cart handle, pushing the cart forward, and pushing a load of *more than 50 kg*. See https://opendrivelab.com/WholeBodyVLA.

## Abstract

Humanoid robots require precise locomotion and dexterous manipulation to perform challenging loco-manipulation tasks. Yet existing approaches, modular or end-to-end, are deficient in manipulation-aware locomotion. This confines the robot to a limited workspace, preventing it from performing large-space loco-manipulation. We attribute this to: (1) the challenge of acquiring loco-manipulation knowledge due to the scarcity of humanoid teleoperation data, and (2) the difficulty of faithfully and reliably executing locomotion commands, stemming from the limited precision and stability of existing RL controllers. To acquire richer loco-manipulation knowledge, we propose a unified latent learning framework that enables Vision-Language-Action (VLA) system to learn from low-cost action-free egocentric videos. Moreover, an efficient human data collection pipeline is devised to augment the dataset and scale the benefits. To execute the desired locomotion commands more precisely, we present a loco–manipulation–oriented (LMO) RL policy specifically tailored for accurate and stable core loco-manipulation movements, such as advancing, turning, and squatting. Building on these components, we introduce WholeBodyVLA, a unified framework for humanoid loco-manipulation. To the best of our knowledge, WholeBodyVLA is one of its kind enabling large-space humanoid loco–manipulation. It is verified via comprehensive experiments on the AgiBot X2 humanoid, outperforming prior baseline by 21.3%. It also demonstrates strong generalization and high extensibility across a broad range of tasks.

# 1 INTRODUCTION

Humanoid robots are widely regarded as the leading embodiment for realizing general-purpose embodied agents—systems that can perceive, reason, and act in the open-ended, human-centered environment. Realizing this vision requires close coordination between dexterous manipulation and agile locomotion. Despite notable progress in RL-based whole-body imitation (Ji et al., 2024; He et al., 2025b;a; Truong et al., 2025) and loco-manipulation controllers (Ben et al., 2025; Zhang et al., 2025a; Li et al., 2025b; Shi et al., 2025), and the recent surge of Vision-Language-Action (VLA) systems for in-place manipulation (Brohan et al., 2023; Kim et al., 2024; Liu et al., 2025; Intelligence et al., 2025; Yu et al., 2026; Yang et al., 2026), autonomous policies for humanoid loco–manipulation remain limited. A key challenge is *manipulation-aware* locomotion: planning and executing movements that actively create the preconditions for the intended manipulation—approaching, orienting, and stabilizing—rather than treating locomotion and manipulation as separate stage.

A naive solution is to serialize locomotion and manipulation with a high-level planner, selecting and switching between them (*e.g.*, navigation vs. grasping) (Yuan et al., 2025; Zhang et al., 2025b; Chen et al., 2025b). However, the limited closed-loop feedback and lack of end-to-end joint optimization could lead to error accumulation, resulting in robot configurations that are suboptimal for subsequent manipulation tasks. Another promising option is via the end-to-end framework (Ding et al., 2025; Bjorck et al., 2025; Boston Dynamics, 2025). It executes whole-body control directly and may alleviate the handoff issues of modular pipelines. However, training such end-to-end policies via imitation learning requires large-scale whole-body data, which are difficult to obtain.

To this end, we contend that the most fundamental concern to overcome is **data scarcity**, which prevents the acquisition of loco-manipulation knowledge. Large-scale datasets have proven critical in tabletop manipulation (Padalkar et al., 2024; Bu et al., 2025a) and in navigation on wheeled or quadruped platforms (Anderson et al., 2018; Ku et al., 2020; Zhang et al., 2026). However, these resources treat manipulation and navigation as separate tasks. In contrast, datasets that integrate humanoid locomotion with manipulation are almost few. Collecting such trajectories at scale, either via Motion Capture (MoCap) or teleoperation, is prohibitively expensive. Without such data, models lack the requisite experience to learn locomotion behaviors that fulfill the manipulation scenarios sufficiently in real-world deployments.

In this work, we explore how to learn loco–manipulation behaviors from low-cost action-free videos, alleviating the scarcity of teleoperation data. Humans naturally acquire new loco–manipulation skills by watching others, and prior work in tabletop manipulation has shown that human demonstration videos can be highly effective (Ye et al., 2025; Bu et al., 2025b; Bjorck et al., 2025). We argue that the same intuition applies to humanoid loco–manipulation: videos already expose key information such as locomotion direction, end-effector trajectories, object affordances, and physical interaction cues. By contrast, teleoperation data offers robot-aligned actions ready for direct imitation learning but require costly hardware and skilled operators, hindering large-scale collection.

Motivated by this gap, we focus on the pre-training stage of VLA and introduce **unified latent learning**, which acquires large-scale loco–manipulation priors from human egocentric videos and uses them as latent supervision for the VLA. Unified latent learning operates by turning action-free videos into discrete latent actions. Since videos lack robot-aligned labels and therefore cannot be used for direct imitation learning, we train a latent action model (LAM) to encode frame-to-frame inverse dynamics into a compact discrete latent space. Because locomotion and manipulation exhibit fundamentally different visual change patterns, we train a locomotion LAM and a manipulation LAM separately. For the locomotion LAM, we use our self-collected egocentric manipulation-aware locomotion videos and design a simple capture pipeline requiring only a single operator with a monocular camera. For the manipulation LAM, we rely on AgiBot World, one of the largest real-robot manipulation datasets. After LAM pre-training, the VLA is trained on mixed human-video and robot-data under joint supervision from both LAMs, ensuring coherent intention prediction across locomotion and manipulation. After pre-training, we attach a lightweight action decoder and fine-tune the VLA on teleoperation trajectories, grounding the latent actions into robot-executable commands: upper-body joint positions and a locomotion command for the lower body. A high-frequency RL controller then converts the locomotion command into reliable lower-body actions.

While unified latent learning provides rich supervision for training the high-level VLA, the robot may still fail at locomotion execution due to limitations of the low-level RL controller. As shown in

Table 1: **Comparison of autonomous humanoid control systems.** We achieve various loco-manipulation tasks in the real world through full-body control, without external modules. Manipulation reflects single- vs. coordinated dual-arm tasks, while Locomotion covers stepping (lateral and forward/backward), turning, and squatting. Information summarized from demos.

| Method | V/L Input | Manipulation | Locomotion | Closed-Loop | Multi-Task | No Extra Info. |
|---|---|---|---|---|---|---|
| HOMIE + IL (Ben et al., 2025) | vision | single arm | ✗ ✗ 🦵↕ | ✓ | ✗ | ✓ |
| AMO + IL (Li et al., 2025a) | vision | single arm | 🚶 ✗ 🦵↕ | ✓ | ✗ | ✓ |
| FALCON + Planner (Zhang et al., 2025a) | vision | dual arm | 🚶 ↻ 🦵↕ | ✓ | ✗ | MoCap input, object pose |
| R²S² (Zhang et al., 2025b) | ✗ | dual arm | 🚶 ✗ 🦵↕ | ✓ | ✗ | MoCap input |
| LeVERB (Xue et al., 2025) | vision, text | ✗ | 🚶 ↻ 🦵↕ | ✓ | ✓ | ✓ |
| HITTER (Su et al., 2025) | ✗ | single arm | 🚶 ✗ 🦵↕ | ✓ | ✗ | MoCap input |
| Humanoid-VLA (Ding et al., 2025) | vision, text | ✗ | 🚶 ✗ ✗ | ✗ | ✗ | ✓ |
| GR00T (Bjorck et al., 2025) | vision, text | dual arm | ✗ ✗ ✗ | ✓ | ✓ | ✓ |
| Being-0 (Yuan et al., 2025) | vision, text | dual arm | 🚶 ✗ ✗ | ✗ | ✓ | GPT4o, detector |
| LBM (Boston Dynamics, 2025) | vision, text | dual arm | ✗ ↻ 🦵↕ | ✓ | ✓ | MoCap data |
| HEAD (Chen et al., 2025b) | vision | ✗ | 🚶 ↻ ✗ | ✓ | ✗ | navigation goal |
| **WholeBodyVLA (Ours)** | vision, text | dual arm | 🚶 ↻ 🦵↕ | ✓ | ✓ | ✓ |

the failure-case statistics in Appendix C.3, many errors—such as *stumble*, *path deviation*, and *turn with advance*—arise from the limited precision and stability of the underlying RL controller rather than from the VLA itself. A key contributor is the continuous velocity-tracking objective used in existing locomotion RL controllers. While suitable for broad locomotion behaviors, this objective exceeds the actual needs of loco-manipulation, making the controller harder to train and less reliable for fine-grained positional control. To address this, we introduce a loco–manipulation–oriented (LMO) RL policy that employs a simplified discrete command interface. This design is specifically tailored for accurate and stable execution of fundamental loco-manipulation movements such as advancing, turning, and squatting, enabling more precise and dependable low-level control.

Based on these designs, we introduce WholeBodyVLA, a framework for efficient training and deployment of autonomous control policies on a bipedal humanoid. It enables real-world, large-space end-to-end humanoid loco–manipulation. As summarized in Table 1, earlier systems either decouple key components or support only partial loco–manipulation, whereas WholeBodyVLA consolidates them within a single unified framework. Comprehensive experiments on the Agibot X2 show that WholeBodyVLA surpasses prior baselines by 21.3% and 24.0%, while demonstrating strong generalization and broad task coverage. Our contributions could be summarized as follows:

- We present **WholeBodyVLA**, a VLA framework that enables the bipedal humanoid to perform end-to-end large-space loco–manipulation in the real-world setting autonomously.
- We introduce **unified latent learning**, enabling joint locomotion–manipulation learning from abundant low-cost action-free videos and alleviating the scarcity of teleoperation data.
- We propose a **loco–manipulation–oriented (LMO)** RL policy that mitigates decision–execution misalignment via a discrete command interface tailored for loco-manipulation.

## 2 RELATED WORKS

### 2.1 HUMANOID WHOLE-BODY CONTROL

**Loco–manipulation controllers.** To move beyond isolated manipulation or motion imitation (Cheng et al., 2024; Ji et al., 2024; He et al., 2024; 2025b;a; Ze et al., 2025; Truong et al., 2025; Chen et al., 2025c; Xie et al., 2025; Wang et al., 2025; Cheng et al., 2025; Fu et al., 2025; Pan et al., 2026), a number of RL-based whole-body controllers have been proposed, most of which adopt a velocity-tracking interface, optimizing per-step errors across commanded speed (Ben et al., 2025; Shi et al., 2025; Li et al., 2025a; Zhang et al., 2025a;b; Sun et al., 2025). While sufficient for cruising, this formulation leaves start–stop semantics implicit, induces fragmented gaits across speed, and provides little supervision for episode-level controllability such as braking precision or heading fidelity—capabilities critical for loco–manipulation. Upper-body influence is typically modeled as task-agnostic noise or motion clips, which sparsely reflect the structured inertial couplings of real tasks (grasp, lift, push), limiting stability under load. Methods like HOMIE's PD-stabilized

arms (Ben et al., 2025), AMO's trajectory-optimization hybrids (Li et al., 2025a), FALCON's force curriculum (Zhang et al., 2025a), $R^2S^2$'s skill libraries (Zhang et al., 2025b), or ULC's unified residual controller (Sun et al., 2025) improve robustness but inherit the limitations of velocity-centric training, yielding inconsistent gaits and unstable teleop-trajectories that hinder low-level stability and high-level VLA policy learning.

**High-level planners for humanoids.** Moreover, RL controllers generally lack the ability to process RGB vision or language inputs directly, they are not sufficient for autonomous task execution. To address this, a complementary line of work has explored high-level planning for humanoids. LE-VERB (Xue et al., 2025) embeds latent verbs into RL for low-level WBC control. Other systems like $R^2S^2$ (Zhang et al., 2025b), Being-0 (Yuan et al., 2025) and HEAD (Chen et al., 2025b) employ modular planners driven by vision-language models (VLMs) that sequence locomotion and manipulation as discrete skills. While conceptually appealing, these frameworks are hindered by brittle skill boundaries—robots often end up in unstable or task-infeasible configurations after locomotion—and by reliance on cloud-based perception, which introduces latency and undermines real-time control. In parallel, initial efforts have attempted to extend Vision-Language-Action frameworks to humanoid robots. For instance, Humanoid-VLA (Ding et al., 2025) focuses on locomotion, while GR00T (Bjorck et al., 2025) targets manipulation for humanoid embodiments; each emphasizes one modality while neglecting the other primitive critical for seamless loco-manipulation task execution. The Boston Dynamics demonstration (Boston Dynamics, 2025) is constrained to a limited workspace, yet relies heavily on expensive MoCap collections of whole-body loco–manipulation. These limitations highlight the need for unified frameworks that couple vision and language with whole-body control, enabling humanoid loco–manipulation without brittle modular boundaries.

## 2.2 VISION-LANGUAGE-ACTION MODELS

Recently, building on multimodal foundation models and imitation learning from large-scale real-robot trajectories, VLA systems have garnered broad attention for their strong generalization and dexterous manipulation capabilities Chen et al. (2025a). Representative efforts include RT-2 (Brohan et al., 2023), OpenVLA (Kim et al., 2024), RDT (Liu et al., 2025), Pi0 (Black et al., 2024), and Pi0.5 (Intelligence et al., 2025). Nevertheless, these models typically emphasize upper-body manipulation only and do not provide a unified, end-to-end solution for the autonomous whole-body control required by loco–manipulation tasks. In contrast, our aim is a unified VLA that integrates locomotion and manipulation, enabling bipedal humanoids to perform loco–manipulation tasks.

**Latent action learning.** Despite rapid progress, current robot datasets remain far smaller than those in the vision and language domains. The core bottleneck is the cost of action-labeled trajectories—expensive teleoperation systems, skilled operators, and substantial collection time. Latent action learning sidesteps this: instead of explicit action labels, it compresses frame-to-frame visual changes into compact, discrete tokens that supervise policy learning from action-free videos. Representative approaches include Genie (Bruce et al., 2024), LAPA (Ye et al., 2025), IGOR (Chen et al., 2024) and UniVLA (Bu et al., 2025b). Collectively, these studies demonstrate that abundant, cross-embodiment, action-free videos can be transformed into effective supervisory signals for VLA training. Motivated by this—and noting that large-scale humanoid loco–manipulation data are even harder to obtain—our work performs latent action learning for locomotion and manipulation in a unified manner, enabling humanoids to perform loco–manipulation with strong generalization.

## 3 WHOLEBODYVLA

WholeBodyVLA, which equips VLA models with locomotion primitives that reliably establish the preconditions for manipulation, leverages **unified latent learning** and a **loco–manipulation–oriented (LMO)** RL policy to enable humanoids to accomplish long-range, large-area tasks. In this section, we detail (i) how unified latent learning contributes to WholeBodyVLA jointly master manipulation and locomotion primitives, and (ii) how our loco–manipulation–oriented RL policy explicitly optimizes stability under dynamic disturbances.

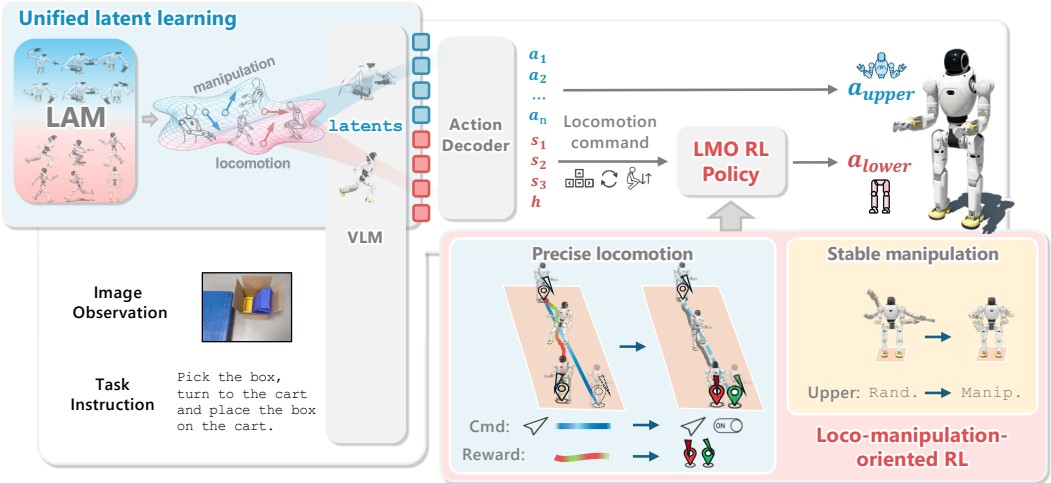

Figure 2: **Pipeline of WholeBodyVLA.** **LAM** is pretrained on manipulation and manipulation-aware locomotion videos, yielding unified latent supervision for the VLM. Meanwhile, the **LMO** RL policy is trained for precise and stable locomotion under disturbances. At runtime, egocentric images and language instructions are encoded by the VLM into latent action tokens, which are decoded ($\sim 10\,\text{Hz}$) into (i) dual-arm joint actions and (ii) locomotion commands executed by **LMO** at $50\,\text{Hz}$, enabling robust whole-body loco–manipulation.

## 3.1 Unified Latent Action Model

Our core idea is to learn manipulation and locomotion primitives from egocentric manipulation and manipulation-aware locomotion videos with Latent Action Models (LAMs), and then supervise VLA training. However, we find that directly training a single LAM on the mixed data yields sub-optimal performance. We attribute this to the fundamentally **different modalities** of the two data sources: in manipulation videos the camera pose is almost static, whereas in locomotion videos it changes continuously. This discrepancy degrades LAM training in two ways. First, image variations in manipulation data are dominated by arm motion, biasing the model to attend to arm regions; conversely, image variations in locomotion data arise mainly from environment motion relative to the moving camera, forcing the model to focus on the entire scene. The conflicting attention objectives hinder stable representation learning. Second, in manipulation data the LAM learns to encode changes in arm-environment relative position as arm motion, whereas in locomotion data (where the arm often remains in the FOV, especially in loco-manipulation tasks) the same relative position changes are caused by camera motion. The LAM may mistakenly interpret these changes as arm motion instead of locomotion, leading to ambiguous latent encodings. Therefore, we train two LAMs separately: a manipulation LAM on the manipulation data and a locomotion LAM on the locomotion data. Both LAMs are then used jointly to supervise the VLA training.

Specifically, following Genie (Bruce et al., 2024) and UniVLA (Bu et al., 2025b), we adopt a VQ-VAE architecture (Van Den Oord et al., 2017) and build the encoder on top of DINOv2 (Oquab et al., 2024) features. Given consecutive frames $(o_t, o_{t+k})$, the LAM encoder $\mathcal{E}$ first emits a continuous latent vector $z_t = \mathcal{E}_i(o_t, o_{t+k})$, which is then quantized to the nearest entry in the learned codebook: $c_t^i = \arg\min_{c \in \mathcal{C}^i} \|z_t - c\|_2$, $c_t \in \mathcal{C}_i$. The index $i \in \{\text{mani}, \text{loco}\}$ denotes the manipulation LAM and the locomotion LAM, respectively. To train both the encoder $\mathcal{E}_i$ and the codebook $\mathcal{C}_i$, the LAM decoder $\mathcal{D}_i$ receives the former frame and the quantized latent action, and is trained to reconstruct the latter frame $\hat{o}_{t+k} = \mathcal{D}_i(o_t, c_t)$. Reconstruction is optimized by minimizing the standard VQ-VAE loss. With the LAMs pretrained, we then train the VLA policy $\pi_\theta$ to jointly predict both types of latent actions given visual observations and task language with Cross-Entropy loss, maximizing

$$\pi_\theta(c_t^{\text{mani}}, c_t^{\text{loco}} \mid o_t, \ell), \tag{1}$$

where $\ell$ represents language instructions. This unified prediction compels the model to learn how locomotion and manipulation interact in a single, cohesive action space to support task execution.

Finally, to execute on humanoids, we introduce a lightweight decoder $f$ that grounds latent actions into robot-specific commands: $a_t = f(\hat{c}_t^{\text{mani}}, \hat{c}_t^{\text{loco}}, s_t)$, where $s_t$ is the robot state. The decoder

produces (i) upper-body joint angles and (ii) a locomotion command indicating the action to be executed. This command is translated by our LMO RL policy to lower-body torques. Through this division of labor, the VLA provides unified latent decisions, the decoder grounds them into embodiment-specific control signals, and the RL policy ensures stable execution—realizing whole-body loco–manipulation in practice.

**Manipulation-aware locomotion data collection.** To further scale the gains of unified latent learning, we propose to exploit large amounts of human egocentric manipulation-aware locomotion videos, which can be easily collected. We design an egocentric data collection pipeline characterized by: (1) *low cost and efficiency*, only a single operator with a head-mounted camera is required, avoiding expensive MoCap or teleoperation; (2) *coverage of humanoid primitives*, operators should perform all types of motion, such as advancing, turning, and squatting; and (3) *goal-directed execution*, operators should perform locomotion to contact potential manipulation goals, ensuring that the locomotion data is directly aligned with loco–manipulation learning. More details, including the framework, training objectives, and data collection pipeline, are provided in Appendix A.1.

## 3.2 LOCO–MANIPULATION–ORIENTED RL POLICY

As discussed in Section 1, a major failure mode in loco–manipulation is the misalignment between high-level decisions and low-level execution. This issue largely arises from the continuous random velocity-tracking objectives used in existing RL controllers—objectives designed for broad locomotion rather than the stable, reliable start–stop and directional control needed for manipulation. In this section, we introduce a **Loco–Manipulation–Oriented (LMO)** RL framework that replaces velocity tracking with a discrete command interface, enabling more faithful execution.

**Observation space.** The policy relies solely on proprioceptive egocentric states with a short history stack: $O_t = [u_t, \boldsymbol{\omega}_t, \mathbf{g}_t, \mathbf{q}_t, \dot{\mathbf{q}}_t, \mathbf{a}_{t-1}]$, including base angular velocity, gravity vector, joint states, and the previous action. This compact design avoids reliance on privileged environment information while remaining sufficient for closed-loop stability.

**Discrete command interface.** We formulate lower-body control as *goal-conditioned regulation*, where the policy executes discrete high-level commands while maintaining balance. At each time step, the planner generates a command $u_t = [s_x, s_y, s_\psi, h^\star] \in \{-1, 0, 1\}^3 \times \mathbb{R}$, where $s_x, s_y, s_\psi$ denotes discrete indicators for forward, lateral, and turning, and $h^\star$ specifies the stance height. Unlike velocity-based formulations, our interface enforces explicit start–stop semantics and reduces trajectory variance, improving the training process of both RL controller and high-level planner.

**Reference shaping.** Since the inputs are ternary flags, we specify a goal speed magnitude $v_k^{\text{goal}} \geq 0$ for each axis, with the sign determined by the intent $s_k$. To avoid abrupt accelerations, directional intents are passed through a smooth gating function:

$$v_k^{\text{ref}}(t) = v_k^{\text{goal}} \tanh\left[\alpha(s_k - \bar{s}_k(t))\right], \qquad \bar{s}_k(t) \leftarrow (1-\lambda)\bar{s}_k(t-1) + \lambda s_k, \qquad (2)$$

for $k \in \{x, y, \psi\}$, where $\bar{s}_k$ is the exponentially smoothed flag. This design ensures predictable on/off transitions and reduces oscillations.

**Two-stage curriculum.** We adopt a two-stage training scheme that first acquires a minimal locomotion skill and then specializes it for precise and stable loco–manipulation.

*Stage I (basic gait acquisition).* For each axis $k \in \{x, y, \psi\}$, if $s_k \neq 0$ we sample a goal speed magnitude $v_k^{\text{goal}} \sim \mathcal{U}([0, v_k^{\max}])$ with the sign determined by $s_k$; otherwise $v_k^{\text{goal}} = 0$. For the upper body, following HOMIE (Ben et al., 2025), the arms track pose targets resampled at a fixed interval and interpolated for smooth motion, while the joint limits are gradually relaxed by a curriculum factor, exposing the legs to progressively stronger disturbances. This stage enables the policy to develop a basic gait that prevents falling, providing a stable foundation for later refinement.

*Stage II (precision and stability).* Stage II further targets loco–manipulation–level precision and stability through dedicated optimization. On the locomotion side, we fix per-axis cruising speed to constants ($v_k^{\text{goal}} = \bar{v}_k$) to standardize cruising and suppress unintended heading drift when no yaw intent is given ($s_\psi = 0$ at onset/offset should not induce yaw). Directional accuracy is measured by the terminal deviation:

$$\mathcal{J}_{\text{dir}} = |\operatorname{wrap}(\psi_{\text{end}} - \psi_{\text{start}})|, \qquad (3)$$

where an episode begins when any axis flag flips $0 \rightarrow \pm 1$ and ends when it returns to $0$ and the base stabilizes. Minimizing $\mathbb{E}[\mathcal{J}_{\text{dir}}]$ enforces precise initiation, steady cruising, and consistent braking. On the manipulation side, we inject realistic perturbations by sampling short arm-motion segments from Agibot-World (Bu et al., 2025a), interpolating them into continuous signals, and replaying them at varied rates with light noise. This forces the legs to compensate for structured inertial couplings rather than unstructured disturbances. Moreover, for stationary episodes ($s_x = s_y = s_\psi = 0$), we add a stand-still penalty to discourage unnecessary leg actions:

$$\mathcal{J}_{\text{stand}} = \|a_i^{\text{leg}}\|_2^2, \tag{4}$$

Additional implementation details are provided in Appendix A.2. Together, these designs yield stable, repeatable gaits and reliable whole-body coordination, avoiding the fragmented motion patterns often induced by velocity-tracking objectives.

# 4 EXPERIMENTS

In this section, we aim to answer four key questions: **Q1.** (Section 4.2) Does WholeBodyVLA enable long-range, large-area loco–manipulation beyond existing SOTA approaches? **Q2.** (Section 4.3) Does learning from action-free videos actually improve performance and reduce reliance on teleoperation data? **Q3.** (Section 4.4) How does LMO contribute to loco–manipulation? **Q4.** (Section 4.5) Does WholeBodyVLA generalize to long-horizon and extended scenarios?

## 4.1 SETUP

**Hardware, tasks, and data collection.** We evaluate on the prototype of Agibot X2 humanoid (7-DoF arms with Omnipicker grippers, 6-DoF legs, 1-DoF waist, and an egocentric Intel RealSense D435i camera). Three tasks comprehensively test loco–manipulation: (i) *bag packing*—grasping a paper bag, sidestepping to a carton, squatting, and placing it inside; (ii) *box loading*—squatting to grasp a box, turning, and placing it onto a cart; and (iii) *cart pushing*—grasping a 50 kg cart handle and pushing it forward stably. These tasks jointly evaluate dual-arm coordination, squat precision, turning accuracy, and stability under heavy loads. Details of the robot platform and teleoperated data collection (VR + joystick, 50 executions per task) are provided in Appendix B.1.

**Baselines.** We compare WholeBodyVLA with representative modular pipelines, including a navigation-assisted *Modular Design* baseline, VLA frameworks GR00T N1.5 (Bjorck et al., 2025) and OpenVLA-OFT (Kim et al., 2025), both adapted to output dual-arm joint actions and the same discrete locomotion command as WholeBodyVLA that is executed by our LMO controller, and ablated variants of our design, which either remove/replace the LMO or modify unified latent learning (no LAM, manipulation-only LAM, or a single shared LAM on mixed data). Full implementation details and training setups are provided in Appendix B.2.

**Training protocol.** We follow the standard VLA recipe of large-scale pretraining followed by real-robot finetuning. For WholeBodyVLA, pretraining has two steps: Stage I pretrains separate manipulation and locomotion LAMs on large egocentric manipulation and manipulation-aware locomotion videos, and Stage II trains the VLA to predict both latent actions on the same corpus using the LAM codes as pseudo-action labels. For VLA baselines (GR00T, OpenVLA-OFT), we use their publicly released pretrained models. In the finetuning stage (Stage III), all methods are trained on the same Agibot-X2 teleoperation trajectories for all tasks. The LMO controller and its velocity-based baseline (Ben et al., 2025) are trained separately in simulation and kept fixed during teleoperation data collection and final deployment.

## 4.2 MAIN RESULTS

To demonstrate the effectiveness of WholeBodyVLA, we design three task suites—each comprising several subgoals and a diverse set of loco–manipulation primitives—to benchmark performance against multiple baselines (Table 2). These suites span a range of real-world challenges: (1) Bag Packing, which requires stable lateral stepping and precise bimanual manipulation; (2) Box Loading, which involves coordinated turning, squatting, and object placement while maintaining balance; and (3) Cart Pushing, which demands sustained forward locomotion and reliable heading control. Across

Table 2: **Evaluation across three tasks.** Each task is decomposed into two subgoals. Whole-BodyVLA is shown to outperform both modular and end-to-end baselines, with unified latent learning and the LMO both contributing significantly.

| Method | Bag Packing | | Box Loading | | Cart Pushing | | Avg. Score |
|---|---|---|---|---|---|---|---|
| | Grasp Bags | Move & Squat | Squat & Grasp | Rise & Turn | Grab Handle | Push Ahead | |
| Modular Design | 22/25 | 12/25 | 9/25 | 9/25 | 22/25 | 22/25 | 64.0% |
| GR00T w/ LMO | 20/25 | 10/25 | 6/25 | 4/25 | 12/25 | 11/25 | 42.0% |
| OpenVLA-OFT w/ LMO | 19/25 | 6/25 | 12/25 | 12/25 | 22/25 | 14/25 | 56.7% |
| **WholeBodyVLA (ours)** | **23/25** | **13/25** | **19/25** | **17/25** | **23/25** | **22/25** | **78.0%** |
| WholeBodyVLA w/o RL | - | - | - | - | - | - | - |
| - w/ vel.-based RL | 22/25 | 1/25 | 16/25 | 3/25 | 24/25 | 15/25 | 54.0% |
| - w/o lam | 15/25 | 4/25 | 8/25 | 6/25 | 16/25 | 10/25 | 39.3% |
| - w/ manip. lam | 24/25 | 7/25 | 17/25 | 11/25 | 20/25 | 14/25 | 63.3% |
| - w/ shared lam | 18/25 | 11/25 | 16/25 | 16/25 | 20/25 | 18/25 | 66.0% |

these tasks, WholeBodyVLA achieves consistently higher success rates than modular and end-to-end baselines, indicating better loco–manipulation behavior in practice.

## 4.3 HOW DOES ACTION-FREE VIDEOS CONTRIBUTE TO LOCO–MANIPULATION?

To address Q2: *Does learning from human egocentric videos actually improve performance and reduce reliance on teleoperation data?*, we compare WholeBodyVLA, which performs full latent pretraining before teleoperation fine-tuning, with WholeBodyVLA w/o LAM, which skips latent pretraining. As shown in Table 2, the full model improves success rate by 38.7%, indicating that unified latent learning extracts useful priors from action-free human videos and enhances downstream policy learning. We also note, in passing, that a shared-LAM variant performs slightly worse than our separate-LAM design, suggesting that decoupling the two LAMs is beneficial but not the primary factor. Additionally, we evaluate a variant that performs latent learning only on in-place manipulation in AgibotWorld (i.e., without locomotion pretraining). The full-pretraining model outperforms this manipulation-only variant by 14.7%, with the largest gains on tasks requiring substantial locomotion before manipulation.

**Data scaling under generalization settings.** To further assess how latent learning improves model performance and reduces reliance on teleoperation data, we conduct targeted generalization experiments under (i) changed start-poses and (ii) changed objects, layouts, and appearance, while keeping the language instruction fixed, see full setups in Appendix C.1. We then study how the amounts of latent pretraining and fine-tuning data affect generalization. In the first group of tasks (Fig. 3 (a)), which primarily evaluate **locomotion generalization**, we compare models pretrained with 0%, 25%, 50%, and 100% human egocentric videos (all using 100% AgibotWorld data). Their average success rates (vertical axis) are plotted against increasing amounts of teleoperation fine-tuning data (horizontal axis). Models pretrained with more human videos consistently perform better. Notably, with more than 50% human video pretraining, the model matches variants using less than 25% human videos even when fine-tuned with only 25 teleoperation trajectories, whereas the latter require 200 trajectories to achieve similar performance. In the second group of tasks (Fig. 3 (b)), focused on **manipulation generalization**, we vary the amount of AgibotWorld data used for latent learning (all using 100% human videos). The trends mirror those in locomotion: stronger latent pretraining yields higher success rates and reduces the amount of fine-tuning data required. Together, these results show that human egocentric videos and unified latent learning significantly improve VLA generalization while reducing reliance on teleoperation data. With a fixed fine-tuning budget, stronger latent pretraining consistently yields higher performance.

## 4.4 HOW DOES LMO CONTRIBUTE TO LOCO–MANIPULATION?

We first compare WholeBodyVLA with a velocity-based RL variant on the task suites in Table 2. Overall, the velocity-based controller achieves a 24% lower success rate, with 91.7% of this gap coming from failures in the second subgoal of each task, which contains most of the locomotion. To further stress-test both controllers, we also evaluate them on the extended tasks in Fig. 3(c), including uneven-terrain traversal, long multi-step sequences, and following extended floor markings

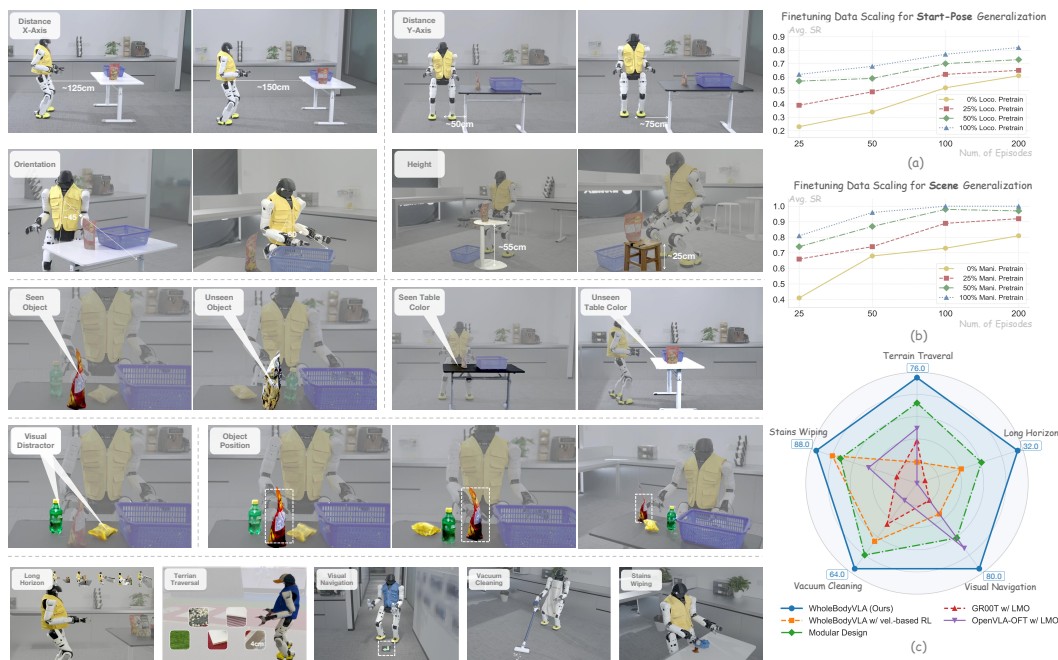

Figure 3: **Real-world generalization of WholeBodyVLA.** Top: variations in robot start-pose and scene appearance, with data-scaling curves. Bottom: comparison on extended tasks with different baselines. See videos on https://opendrivelab.com/WholeBodyVLA.

for visual navigation—settings dominated by locomotion and long-horizon execution. Across these scenarios, the velocity-based RL variant fails substantially more often than WholeBodyVLA. These failures largely arise from the suboptimal behavior of the vanilla velocity-based controller, which often produces errors such as stumbling, path deviation, or turning while advancing, independent of the high-level VLA decisions. A detailed analysis is provided in Appendix C.3. Taken together, these observations indicate that our LMO RL policy effectively mitigates this issue—one that may appear minor but is critical for reliable long-range and multi-step loco–manipulation.

Table 3: **Ablations of the LMO design under locomotion accuracy and manipulation stability.** Locomotion accuracy is evaluated for forward/backward walking, lateral stepping, and turning, reported as position/orientation error (mean $\pm$ std). Manipulation stability is quantified by CoM sway during standing and squatting; lower is better. See details in Appendix B.4.

| Method | Locomotion Accuracy (Pos. / Quat. Error) | | | Manipulation Stability (CoMS) | |
|---|---|---|---|---|---|
| | Forward&Backward | Left&Right | Turning | Standing | Squatting |
| **LMO (ours)** | **0.21±0.01 / 0.05±0.01** | **0.55±0.01 / 0.06±0.01** | **0.05±0.01 / 0.19±0.01** | **0.03±0.02** | **0.03±0.02** |
| LMO w/o Eq. 3 | 0.24±0.02 / 0.07±0.01 | 0.61±0.02 / 0.09±0.01 | 0.05±0.01 / 0.28±0.02 | 0.04±0.03 | 0.03±0.02 |
| LMO w/o stage 2 | 0.27±0.02 / 0.09±0.01 | 0.72±0.03 / 0.11±0.02 | 0.20±0.01 / 0.32±0.03 | 0.05±0.04 | 0.07±0.03 |
| LMO w/o stage 1 | 0.30±0.03 / 0.11±0.01 | 0.66±0.04 / 0.13±0.03 | 0.46±0.01 / 0.34±0.04 | 0.05±0.03 | 0.04±0.03 |
| Vel.-based policy | 0.24±0.04 / 0.12±0.02 | 0.60±0.05 / 0.17±0.06 | 0.26±0.01 / 0.20±0.06 | 0.06±0.04 | 0.05±0.04 |

Beyond real-robot results, we perform controlled ablations in MuJoCo as shown in Table 3. Removing the directional accuracy reward (w/o Eq. 3) degrades turning precision. Disabling Stage II increases trajectory error and squatting sway, showing the necessity of targeted refinement. Without Stage I, the policy fails to acquire stable gaits, producing the largest errors overall. These results confirm that the discrete command interface, two-stage curriculum, and structured perturbations are all crucial for precise trajectory tracking and stable whole-body coordination.

### 4.5 DOES WHOLEBODYVLA GENERALIZE TO LONG-HORIZON AND EXTENDED LOCO-MANIPULATION SCENARIOS?

We also evaluate the extensibility of WholeBodyVLA by evaluating it on a set of more challenging scenarios, detailed in Appendix C.1 The bottom row of Fig. 3 shows five extended tasks, including traversing uneven terrain, executing a long-horizon multi-step sequence, following floor markings for visual navigation, and everyday loco–manipulation activities such as wiping a table and vacuum cleaning. As shown in Fig. 3 (c), WholeBodyVLA remains superior across all these settings, indicating the framework scales beyond the benchmark tasks while preserving robust generalization.

## 5 CONCLUSION AND FUTURE WORK

We present WholeBodyVLA, a VLA enabling humanoid robots to perform large-space loco–manipulation. WholeBodyVLA introduces unified latent learning and a LMO RL policy, effectively alleviating the scarcity of teleoperation data as well as the decision-execution misalignment induced by redundant velocity-tracking objectives in prior RL training. Comprehensive experiments show superior performance, strong generalization and extensibility compared to prior baselines.

**Limitations and future work.** While effective, WholeBodyVLA still faces challenges in handling long-horizon and dexterous tasks. Future work will focus on incorporating lightweight mapping and memory for extended planning and developing active perception strategies to improve robustness in cluttered or dynamic environments. These directions will further enhance its scalability and generalization, paving the way toward versatile real-world humanoid loco–manipulation.

## ACKNOWLEDGMENT

This work is supported by the JC STEM Lab of Autonomous Intelligent Systems funded by The Hong Kong Jockey Club Charities Trust. We also gratefully acknowledge the hardware modification support and generous research sponsorship from AGIBOT. We would like to express our sincere gratitude to Hanfu Gai, Yixuan Pan, Yichao Zhong, Chuanchun Lin, Xiangchao Shi, Kun Wang, Minghao Zhang, Hexing Ai and the rest of the members from OpenDriveLab and Agibot X-Lab for their discussions and support throughout this work. We would also like to thank Xingge Qiao, Ziqi Dai and Zhihao Sun for their assistance with video shooting.

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

## *Appendix*

## A METHODOLOGY

### A.1 DETAILS OF LAM AND VLA ARCHITECTURE

This section provides additional details of our VLA, complementing Section 3.1.

**LAM training.** The LAM encoder $\mathcal{E}_i$ is implemented with a spatio-temporal transformer. Given consecutive frames $(o_t, o_{t+k})$, it generates a continuous latent vector:

$$z_t = \mathcal{E}_i(o_t, o_{t+k}),$$

which is then quantized to the closest entry in the learned codebook:

$$c_t = \text{quantize}(z_t) := \arg\min_{c \in \mathcal{C}} |z_t - c|_2, \qquad c_t \in \mathcal{C}_i.$$

Here, $i \in \{\text{mani}, \text{loco}\}$ denotes whether the LAM corresponds to manipulation or locomotion.

To optimize both the encoder $\mathcal{E}_i$ and the codebook $\mathcal{C}i$, the decoder $\mathcal{D}i$ reconstructs the future frame $\hat{o}t + k$ from the current frame $o_t$ and the quantized latent action $c_t$:

$$\hat{o}_{t+k} = \mathcal{D}_i(o_t, c_t).$$

Reconstruction is supervised with a mean-squared error loss:

$$\mathcal{L}_{\text{mse}} = |o_{t+k} - \hat{o}_{t+k}|_2^2.$$

In addition, we employ the standard VQ-VAE objective to jointly train the encoder, decoder, and codebook:

$$\mathcal{L}_{\text{LAM}} = \mathcal{L}_{\text{mse}} + \|\text{sg}[c_t] - z_t\|_2^2 + \beta \|c_t - \text{sg}[z_t]\|_2^2,$$

where $\text{sg}[\cdot]$ indicates the stop-gradient operator and $\beta$ is the commitment cost.

**VLA training.** After pretraining the LAMs, we train the vision-language-action (VLA) policy $\pi_\theta$ to predict both manipulation and locomotion latent actions from visual observations and task language. This is formulated as maximum likelihood estimation (MLE):

$$\min_\theta [-\log \pi_\theta(c_t^{\text{mani}}, c_t^{\text{loco}} \mid o_t, \ell)].$$

This joint prediction forces the policy to model the interaction of locomotion and manipulation in a unified latent space for task execution.

**Execution on humanoids.** For deployment, we use a lightweight execution decoder $f$ to map latent actions into robot-specific control commands:

$$a_t = f(\hat{c}_t^{\text{mani}}, \hat{c}_t^{\text{loco}}, s_t),$$

where $s_t$ is the robot state, and $\hat{c}_t^{\text{mani}}, \hat{c}_t^{\text{loco}}$ are the latent actions predicted by the VLA policy. The decoder produces two outputs: (1) upper-body joint angles for manipulation, and (2) a locomotion command for the low-level RL controller.

**Manipulation-aware locomotion data collection details.** We design a simple yet effective data collection pipeline that requires only a single operator wearing a camera to capture data. During collection, we employ two types of cameras: (1) an Intel RealSense D435i RGB-D camera, and (2) a GoPro camera, which provides a larger field of view (FOV) and thus enables the collection of more suitable locomotion data for model learning. The operator mounts the camera on the head and performs a variety of loco-manipulation tasks. For the manipulation component, the operator is not required to actually execute object interactions; instead, it suffices to identify potentially manipulable objects and approach them until contact. For locomotion, we instruct the operator to perform diverse actions such as advancing, turning, and squatting. A schematic overview of our data collection pipeline is shown in Figure 4.

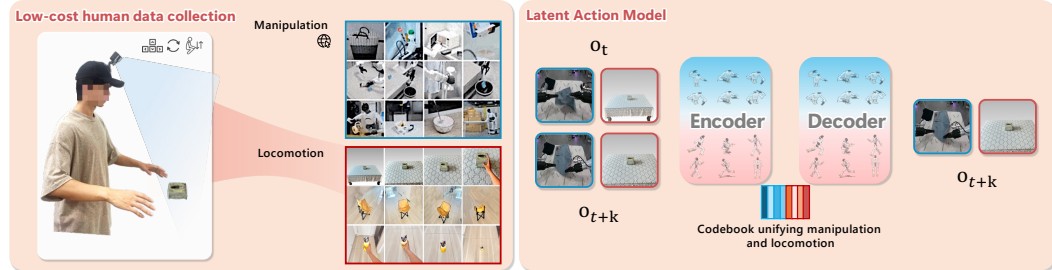

Figure 4: **Low-cost egocentric data collection on locomotion task and pretraining pipeline for LAM.** A single operator records egocentric video while executing eight canonical motion primitives toward potential manipulation goals. This task-oriented pipeline captures locomotion patterns that are diverse, structured, and directly relevant to humanoid loco–manipulation. Then the collected manipulation-aware locomotion data along with the open-source in-place manipulation dataset are used to pretrain the latent action model in a VQ-VAE-style pipeline.

## A.2 DETAILS OF THE LOCO–MANIPULATION–ORIENTED RL POLICY

This section provides additional details of our Loco–Manipulation–Oriented (LMO) RL policy, complementing Section 3.2. Unlike conventional velocity-tracking controllers, which often produce inconsistent gaits when tracking different reference speed, our policy operates in a discrete intent space (*start*, *stop*, *turn*, *squat*) that aligns more naturally with loco–manipulation tasks. The following subsections describe the control interface, policy inputs, reference shaping, curriculum design, and additional implementation details.

**Problem formulation.** We cast lower-body control as *goal-conditioned regulation*, where the objective is to execute discrete commands faithfully while maintaining balance under manipulation. At each step $t$, the planner issues

$$u_t = [\, s_x, s_y, s_\psi, h^\star \,] \in \{-1, 0, 1\}^3 \times \mathbb{R},$$

where $s.$ are discrete start/stop flags for forward/backward ($s_x$), lateral ($s_y$), and yaw rotation ($s_\psi$), and $h^\star$ is the desired stance height. This command interface provides explicit semantics for start–stop execution, in contrast to velocity-based objectives that yield inconsistent or unstable gaits when trained across varying reference speed.

**Observation space.** The policy receives purely proprioceptive inputs with a short history stack:

$$O_t = [\, u_t, \, \boldsymbol{\omega}_t, \, \mathbf{g}_t, \, \mathbf{q}_t, \, \dot{\mathbf{q}}_t, \, \mathbf{a}_{t-1} \,],$$

where $\boldsymbol{\omega}_t \in \mathbb{R}^3$ is the base angular velocity, $\mathbf{g}_t \in \mathbb{R}^3$ the gravity vector, $\mathbf{q}_t, \dot{\mathbf{q}}_t$ the joint positions and velocities, and $\mathbf{a}_{t-1}$ the previous action. This design avoids reliance on privileged simulator information and is sufficient for closed-loop balance.

**Reference shaping.** Because intents $s_k$ are ternary, we convert them into smooth velocity references to prevent impulsive accelerations:

$$v_k^{\text{ref}}(t) = v_k^{\text{goal}} \sigma\big(\alpha(s_k - \bar{s}_k(t))\big), \qquad \bar{s}_k(t) \leftarrow (1 - \lambda)\, \bar{s}_k(t-1) + \lambda\, s_k,$$

where $v_k^{\text{goal}}$ is the goal speed magnitude, $\sigma(\cdot)$ a saturating nonlinearity, and $\bar{s}_k$ the smoothed flag. This "soft gate" ensures predictable transitions when toggling intents on or off, implicitly bounding accelerations.

**Two-stage curriculum.** *Stage I (basic gait acquisition).* In the first stage, the policy learns a minimal gait that prevents falls and responds to discrete commands. For each axis $k$, if $s_k \neq 0$ we sample $v_k^{\text{goal}} \sim \mathcal{U}([0, v_k^{\text{max}}])$, otherwise $v_k^{\text{goal}} = 0$. The upper body follows simple pose targets that are resampled at a fixed cadence, while joint ranges are gradually expanded by a curriculum factor. This exposes the legs to progressively stronger but task-agnostic disturbances, establishing a stable baseline gait.

*Stage II (precision and stability).* In the second stage, the focus shifts to controllability and robustness under manipulation. Cruising speed are fixed ($v_k^{\text{goal}} = \bar{v}_k$) to prevent fragmented gaits. Directional accuracy is enforced by penalizing yaw drift at episode termination:

$$\mathcal{J}_{\text{dir}} = |\operatorname{wrap}(\psi_{\text{end}} - \psi_{\text{start}})|,$$

where episodes are defined between flag flips $0 \leftrightarrow \pm 1$. Structured perturbations are introduced by replaying short arm-motion clips from Agibot-World (Bu et al., 2025a), interpolated and time-warped:

$$\omega_{i+1} = \min(L, \omega_i + (\gamma + \delta_i)\Delta t), \qquad \omega_0 = 0,$$

with $L \sim \operatorname{Unif}[0.8, 2.5]$, $\gamma \sim \operatorname{Unif}[0.8, 1.5]$, and $\delta_i \sim \operatorname{Unif}[-0.25, 0.25]$. Per-step targets are given by $q_i^{\text{tar}} = q^{\text{arm}}(\omega_i) + \varepsilon_i$ with $\varepsilon_i \sim \mathcal{N}(0, 0.05^2)$. This structured disturbance forces the legs to compensate for realistic inertial coupling, unlike random perturbations. Additionally, for stationary episodes ($s_x = s_y = s_\psi = 0$) we apply a stand-still penalty

$$\mathcal{J}_{\text{stand}} = \|a_i^{\text{leg}}\|_2^2,$$

discouraging spurious leg motions and ensuring balance.

**Reward functions and domain randomization.** Table 4 summarizes all reward terms and weights, separated by category (intent execution, posture, locomotion structure, smoothness, and stability). Novel terms such as yaw-drift penalties, structured perturbation compensation, and stand-still regularization are introduced in Stage II to improve loco–manipulation robustness.

Table 5 details the domain randomization parameters. We randomize dynamics (joint torque injection, actuation offsets, link mass), contact properties (friction, restitution, payload), controller gains ($K_p, K_d$), and sensory delays (DOF/IMU lag). Stage II increases the strength and frequency of perturbations, including push disturbances and manipulation-induced payload variations, to further enhance robustness.

## B   TRAINING AND DEPLOYMENT DETAILS

For LAM and VLA training, locomotion LAM is trained on our collected low-cost egocentric locomotion videos, while manipulation LAM is trained from real robot bimanual manipulation datasets. For locomotion data, we collect about 300 hours covering various scenes with our data collection pipeline. For manipulation data, we use AgiBot World (Bu et al., 2025a) dataset. For the shared LAM mentioned in Section 4.3, we train the model on mixed data. Because the two data sources are imbalanced in size, we perform balanced sampling during training from both sources in every batch. For LAM training, we fix the training schedule to 30,000 steps, using a total batch size of 256. For VLA training, we train from Prismatic-7B and fix the training schedule to 20,000 steps, using a total batch size of 1024. We then finetune the VLA with LoRA (Hu et al., 2022) on task-specific data. Notably, for the experiments in 4.2, we finetune one model on all three tasks, as opposed to task-specific finetuning. The finetuning process is conducted with a total batch size of 64 and lasts for 10,000 steps. All of our LAM and VLA training are performed on 8×NVIDIA H100 GPUs, while our RL policy is trained on a single NVIDIA H100.

At deployment, the VLA runs on an RTX 4090 GPU workstation, while the RL policy is deployed on a NanoPi onboard computer. Communication between the VLA and the robot is handled via ZeroMQ over Ethernet, enabling low-latency command streaming for closed-loop loco–manipulation control. The LMO policy runs at 50 Hz on proprioceptive inputs, while the VLA backbone operates at ~10 Hz for perception and reasoning.

### B.1   DATA COLLECTION AND HARDWARE DETAILS

**Task design.** We design three task suites that comprehensively evaluate loco–manipulation capabilities by combining various locomotion primitives (forward/backward walking, lateral stepping, in-place turning, squatting) with manipulation actions requiring single-arm or dual-arm coordination. The first task, *bag packing*, requires the robot to grasp a paper bag on the table with both arms, sidestep toward a nearby carton, squat down, and place the bag inside. This task emphasizes

Table 4: Reward functions and weights used to train the LMO policy. Stage I and Stage II currently share the same scales.

| Category | Reward Term | Equation (sketch) | Stage I | Stage II |
|---|---|---|---|---|
| **Intent Execution** | Forward intent execution | $\exp\{-4\,(v_x - s_x \cdot v_x^{\text{goal}})^2\}$ | 1.5 | 1.8 |
| | Lateral intent execution | $\exp\{-4\,(v_y - s_y \cdot v_y^{\text{goal}})^2\}$ | 1.0 | 1.2 |
| | Yaw intent execution | $\exp\{-4\,(\omega_{\text{yaw}} - s_\psi \cdot v_\psi^{\text{goal}})^2\}$ | 2.0 | 2.0 |
| | height tracking | $\exp\{-4\,(h_t - h_{r,t})^2\}$ | 2.0 | 2.0 |
| | Vertical velocity suppression | $v_{r,z}$ | -0.5 | -0.75 |
| | Angular vel. xy penalty | $\|\omega_{r,xy}\|^2$ | -0.025 | -0.05 |
| **Posture & Joints** | Roll/pitch stabilization | $\|\mathbf{g}_x\|^2 + \|\mathbf{g}_y\|^2$ | -1.5 | -1.5 |
| | Hip joint deviation | $\|\theta_{\text{hip}} - \theta^{default}\|^2$ | -0.2 | -0.2 |
| | Ankle joint deviation | $\|\theta_{\text{ankle}} - \theta^{default}\|^2$ | -0.5 | -0.5 |
| | Knee deviation (squat) | penalty on knee angle / height | -0.75 | -0.75 |
| | DoF acceleration penalty | $\|\dot{q}_i - \dot{q}_{i-1}\|^2/dt$ | $-2.5 \times 10^{-7}$ | $-2.5 \times 10^{-7}$ |
| | DoF pos. limit violation | $\sum out_i$ | -2.0 | -2.0 |
| | DoF velocity penalty | $\sum \dot{\theta}_i^2$ | $-1 \times 10^{-4}$ | $-1 \times 10^{-4}$ |
| | DoF velocity limit violation | $\sum RELU(\dot{\theta}_i - \dot{\theta}_i^{max})$ | -0.002 | -0.002 |
| | Torque limit violation | $\sum RELU(\tau_i - \tau_i^{max})$ | -0.1 | -0.1 |
| **Locomotion Structure** | Feet air time | $\mathbf{1}^{(first\ contact)}(T_{air} - 0.5)$ | 0.05 | 0.05 |
| | Foot clearance | $(p_z^{target} - p_z^i)^2 \cdot \dot{v}_{xy}^i$ | -0.25 | -0.25 |
| | Foot lateral spacing | $|y_L^B - y_R^B| - d_{min}$ | 0.5 | 0.5 |
| | Knee lateral spacing | $|y_{\text{left knee}}^B - y_{\text{right knee}}^B| - d_{min}$ | 1.0 | 1.0 |
| | Feet ground parallelism | $\sum Var(H_i)$ | -2.0 | -2.0 |
| | Feet parallelism | $Var(D)$ | -3.0 | -3.0 |
| | No-fly penalty | $\mathbf{1}\{\text{only one foot on ground}\}$ | 0.75 | 0.75 |
| | Foot slip | $|v^{foot}| \cdot \mathbf{1}_{new\ contact}$ | -0.25 | -0.25 |
| | Foot stumble | $\mathbf{1}\{|F^x| > 3|F^z|\}$ | -1.5 | -1.5 |
| **Energy & Smoothness** | Action rate penalty | $\|\mathbf{a}_t - \mathbf{a}_{t-1}\|^2$ | -0.01 | -0.01 |
| | Smoothness (2nd-order) | $\|\mathbf{a}_t - 2\mathbf{a}_{t-1} + \mathbf{a}_{t-2}\|^2$ | -0.05 | -0.05 |
| | Joint power | $\|\mathbf{w}_i\|^2 + 0.2\|\mathbf{w}_i\|^2$ | $-2.0 \times 10^{-5}$ | $-2.0 \times 10^{-5}$ |
| | Torque usage | $\sum \tau_i^2/k_i^p$ | $-2.5 \times 10^{-6}$ | $-2.5 \times 10^{-6}$ |
| | Feet contact force | $\sum RELU(F_i^z - F_{th})$ | -0.00025 | -0.00025 |
| | Contact momentum | $\sum |v_i^z \cdot F_i^z|$ | 0.00025 | 0.00025 |
| | Action vanish penalty | $\max(0, a_i - a_{max}) + ...$ | -1.0 | -1.0 |
| | Roll action zero penalty | roll actuator $\rightarrow 0$ | -0.05 | -0.1 |
| **Stability** | Stand-still penalty | $\|a_i^{leg}\|^2 \cdot \mathbf{1}_{s_x = s_y = s_\psi = 0}$ | -0.05 | -0.1 |
| | Joint tracking error | $\|\theta_i - \theta_i^{target}\|^2$ | -0.1 | -0.1 |

coordinated dual-arm grasping, lateral stepping, and precise squat execution aligned with the placement target. The second task, *box loading*, requires the robot to squat, grasp a box with both hands, stand up, turn to face a cart, and place the box. This stresses squat control, dual-arm stability, and turning accuracy to ensure successful placement. The third task, *cart pushing*, requires the robot to grasp the handle of a 50 kg cart with both arms and push it several meters forward without lateral drift, testing the ability to maintain whole-body stability under sustained external load. These tasks evaluate dual-arm coordination, squat stability, turning precision, and robustness to heavy loads.

**Hardware and real-robot loco-manipulation data collection.** All experiments are conducted on the prototype Agibot X2 humanoid platform (Fig. 5). Each arm has 7 DoF and is equipped with an Omnipicker gripper as the end-effector, the waist provides 1 DoF, and each leg has 6 DoF, supporting a wide range of whole-body motions. For egocentric perception, we mount an Intel RealSense D435i RGB-D camera on the head, which provides synchronized RGB streams for both locomotion and manipulation tasks. We collect training and evaluation data via physical robot teleoperation. A Meta Quest Pro headset provides egocentric VR teleoperation of the upper body, while locomotion

Table 5: Domain randomization parameters used in training.

| Category | Parameter | Range / Setting |
|---|---|---|
| **Dynamics** | Joint torque injection | $[-0.05,\ 0.05]$ |
| | Actuation offset | $[-0.05,\ 0.05]$ |
| | Link mass scale | $[0.8,\ 1.2]$ |
| | COM displacement | $[-0.1,\ 0.1]$ |
| | Body displacement | $[-0.1,\ 0.1]$ |
| **Contact Properties** | Friction coefficient | $[0.1,\ 3.0]$ |
| | Restitution coefficient | $[0.0,\ 1.0]$ |
| | Payload mass (torso) | $[-5,\ 10]$ |
| | Payload mass (hands) | $[-0.1,\ 0.3]$ |
| **Controller Gains** | PD gain scaling $K_p$ | $[0.9,\ 1.1]$ |
| | PD gain scaling $K_d$ | $[0.9,\ 1.1]$ |
| | Initial joint position scale | $[0.8,\ 1.2]$ |
| **External Disturbances** | Push disturbances | velocity up to 0.5 m/s |
| | Push interval | every 4 s |
| | Init upper-body ratio | 0.0 |
| | Delay flag | enabled |
| **Latency / Sensor Noise** | Action lag timesteps | $[2,\ 8]$ |
| | DOF state lag timesteps | $[0,\ 8]$ |
| | IMU lag timesteps | $[1,\ 10]$ |

commands are issued through a joystick. Each of the three tasks is executed 50 times to obtain diverse trajectories for training.

## B.2 BASELINES AND ABLATIONS

**Modular Design.** To emulate a modular pipeline, we replace the navigation module with a human teleoperator wearing an FPV headset. The operator controls only locomotion via a handheld joystick, without exposure to our data-collection process and relying solely on task instructions and scene context. Once navigation ends, control is handed over to WholeBodyVLA for manipulation, during which the operator is fully disabled; control returns to the operator after manipulation completes. This ensures a fair comparison and provides a near-oracle upper bound for modular pipelines.

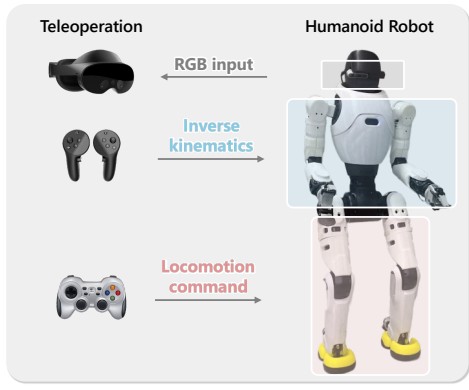

Figure 5: **Description of the hardware.** VR and a joystick are used for data collection.

**GR00T w/ LMO.** GR00T N1.5 (Bjorck et al., 2025) is a recent VLA enabling whole-body control. For fairness, we adapt its output: instead of directly predicting lower-body joints, GR00T predicts locomotion commands, which are then executed by our LMO controller. This isolates the high-level reasoning capacity of GR00T from the low-level locomotion stability issues.

**OpenVLA-OFT w/ LMO.** Since architectural and model-size differences may confound the comparison with GR00T, we further evaluate OpenVLA-OFT (Kim et al., 2025), which shares the same Prismatic-7B initialization as WholeBodyVLA. OpenVLA-OFT is trained to predict upper-body joint actions and locomotion commands, executed by the same LMO controller as in our system.

**Ablations of WholeBodyVLA.** We include five variants to assess the contribution of each component:

(a) **WholeBodyVLA w/o RL**: the VLA directly predicts lower-body joints without the LMO policy.

(b) **WholeBodyVLA w/ Velocity-Based RL**: replacing our LMO with a conventional velocity-tracking RL controller (reproduced and refined from HOMIE (Ben et al., 2025)).

(c) **WholeBodyVLA w/o LAM**: the VLA is directly finetuned from Prismatic-7B without unified latent learning.

(d) **WholeBodyVLA w/ Manipulation LAM**: trained only with manipulation latent learning, without locomotion-aware pretraining.

(e) **WholeBodyVLA w/ Shared LAM**: unified latent learning is performed on mixed data without modality separation.

These baselines together span modular, end-to-end, and ablated configurations, providing a comprehensive evaluation of WholeBodyVLA 's design choices.

## B.3 PROTOCOLS FOR REAL-ROBOT EXPERIMENT

To ensure a fair and reproducible comparison, we adopt the following evaluation protocol: (i) Two independent judges—naive to our data-collection process—adjudicate success/failure for each subgoal and reconcile to a consensus label. Method order is randomized, and judges are blinded to the active policy to mitigate subjective variability. (ii) For each task, subgoals are evaluated sequentially: if the first subgoal fails, the second is automatically counted as a failure. Each task is evaluated over 25 trials, and we report the mean score across all subgoals.

## B.4 DETAILS OF SIMULATION SETUP IN SECTION 4.4

We evaluate full-body humanoid control in MuJoCo using the X2 model under a fixed simulation time-step and control frequency (as specified by `simulation_dt` and `control_decimation`) to ensure comparability across policies. Two complementary experiments are considered: (i) locomotion tracking accuracy and (ii) in-place manipulation stability.

**Locomotion accuracy.** We test three canonical primitives: forward/backward walking ($|v_x| = 0.3\,$m/s), lateral stepping ($|v_y| = 0.3\,$m/s), and in-place turning ($|w_z| = 0.3\,$rad/s). Each trial consists of a 5 s *active phase* with a constant command, followed by a 10 s *settling phase* with zero commands. A reference pose is obtained by integrating the commanded velocity only during the active phase. Metrics are computed exclusively in the settling phase relative to this reference, capturing the controller's stop-and-settle precision. We report the mean $\pm$ standard deviation of (i) position error (m) and (ii) yaw orientation error (rad).

**Manipulation stability.** We evaluate two postures: standing and squatting (achieved by lowering the stance height). During the test, 14-DoF upper-body trajectories from `aligned_joints.h5` are replayed, mapped to the robot's URDF joint order. To amplify coupling effects, trajectories are scaled to $2.0\times$ speed, $1.5\times$ amplitude, and perturbed with Gaussian noise ($\sigma = 0.02$). In addition, randomized external forces are applied: horizontal pushes (up to 150 N) and yaw torques (up to 30 Nm), each lasting 0.2 s and injected approximately every 2.5 s with temporal jitter. Stability is quantified by the *Center-of-Mass Sway* (CoMS, m), defined as the RMS deviation of the horizontal CoM projection:

$$\text{CoM Sway} = \sqrt{\tfrac{1}{T} \int_0^T \|\mathbf{c}(t) - \bar{\mathbf{c}}\|^2 \, dt}, \tag{5}$$

where $\mathbf{c}(t) \in \mathbb{R}^2$ is the horizontal CoM trajectory and $\bar{\mathbf{c}}$ its temporal mean. Lower CoMS indicates improved balance. We report mean $\pm$ standard deviation across trials.

**Control and observations.** Legs are actuated through PD torque control using policy outputs mapped to URDF joints. During stability experiments, the arms follow replayed trajectories; other-

wise, they remain near rest. Observation construction and joint ordering are consistent with training to avoid distribution shift.

# C   MORE EXPERIMENTS

## C.1   GENERALIZATION EXPERIMENT DETAILS

We conduct 12 generalization experiments to evaluate how model performance varies under different start configurations, scene variations, and task settings (Fig. 3). The detailed setups for all experiments are provided below.

The first group concerns start-pose generalization, where only the robot's initial configuration varies while the table, snack bag, basket, and their relative placements remain fixed unless specified.

(1) Distance (X-Axis). The robot starts at different distances from the table and must approach and place a snack bag into the basket. During data collection, start-poses were uniformly sampled between 1.0 m and 1.25 m. For evaluation, start positions of 1.0 m, 1.25 m, and 1.5 m were used (10 trials each). On odd-numbered trials the table appearance matched training, and on even-numbered trials the table color was changed.

(2) Distance (Y-Axis). The robot starts at lateral offsets along the Y-axis. During data collection, start-poses were sampled uniformly between 25 cm and 50 cm on both sides of the table. Evaluation used offsets of 25 cm, 50 cm, and 75 cm on both sides (10 trials per condition, 60 total), with the table recolored on even-numbered trials.

(3) Orientation. The robot begins with different initial headings before rotating toward the table and placing the snack bag. During data collection, orientations were uniformly sampled between $\pm 30°$ and $\pm 60°$. Evaluation tested $\pm 30°$, $\pm 45°$, and $\pm 60°$ (10 trials each), again alternating table appearance across trials.

(4) Height. The snack bag is placed on tables of different heights while the basket remains on the ground. During data collection, three table heights—60 cm, 45 cm, and 25 cm—were randomly chosen. Evaluation included these three heights as well as three unseen heights (55 cm, 40 cm, 20 cm), each tested 10 times.

The second group addresses scene generalization, where visual appearance or object arrangement varies while the underlying task remains fixed.

(5) Unseen Object. The robot manipulates a snack bag not observed during training. During data collection, two designated snack bags are selected randomly each demonstration, with distractor objects on the table. Evaluation involved two novel snack bags, tested over 10 trials each, and all object positions matched those used during data collection.

(6) Unseen Table. The robot performs the same manipulation task on a table with unseen appearance. During data collection, the table, snack bag, basket, and distractors were fixed. Evaluation replaced the table with a visually distinct one while keeping all positions unchanged, and 10 trials were conducted.

(7) Unseen Object Position. The snack bag appears at positions outside the training distribution. During data collection, the bag was randomized uniformly within a 30 cm $\times$ 30 cm region on the table. Evaluation tested two positions lying outside this region (10 trials per position, 20 total).

The final group covers additional locomotion and long-horizon tasks that require more complex behaviors.

(8) Terrain Traversal. The robot receives a bag from a person, follows the person across five terrain types—steps ($\tilde{5}$ cm), foam, wooden planks, gravel, and artificial turf—while maintaining balance, and returns the bag at the end. We collected 50 demonstrations and evaluated 25 trials.

(9) Long-Horizon Manipulation. The robot walks to a table, grasps two specific snack bags, walks to a drawer, places the bags inside, and closes the drawer. 200 demonstrations were collected. Evaluation always used a 1 m start distance and identical object placement to the training setting.

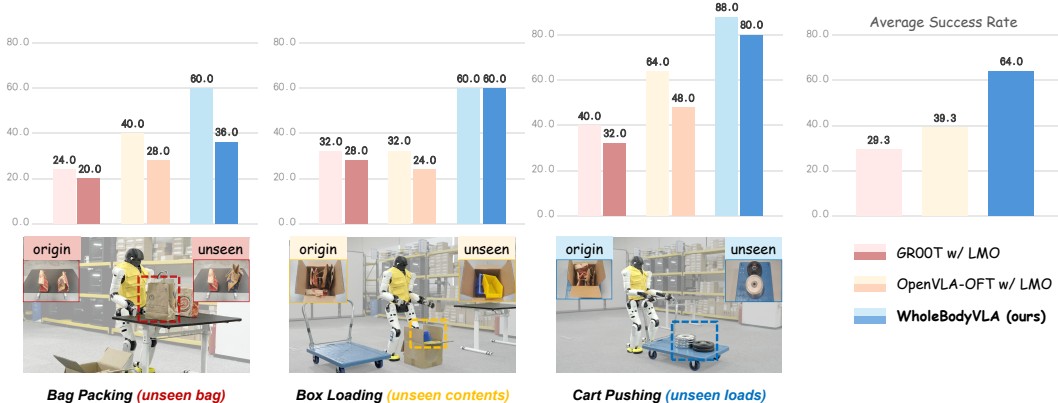

Figure 6: **Visual generalization under object and load variations.** We evaluate Bag Packing, Box Loading, and Cart Pushing under unseen variations of bag appearance, box contents, and cart loads. WholeBodyVLA consistently outperforms GR00T and OpenVLA-OFT across all three tasks, indicating robustness to these distribution shifts.

(10) Visual Navigation. The robot follows a trajectory indicated by visual arrow markers on the ground. We collected 50 demonstrations from a fixed start-pose and evaluated 25 trials under the same configuration.

(11) Vacuum Cleaning. The robot uses a vacuum cleaner to remove paper debris scattered on the floor. Both the start-pose and debris region were fixed during data collection (50 demonstrations) and evaluation (25 trials).

(12) Wiping Stains. The robot wipes coffee stains from a tabletop using a cloth. We collected 100 demonstrations, with stain positions uniformly randomized within a 30 cm × 30 cm region and the cloth placement fixed. During evaluation, the same region and cloth configuration were used (25 trials).

Success rates for all tasks are reported in Fig. 3. The panel titled "Finetuning Data Scaling for Start-Pose Generalization" presents averages over tasks (1)–(4). "Finetuning Data Scaling for Scene Generalization" summarizes tasks (5)–(7). The radar chart reports performance on tasks (8)–(12).

### C.2 ADDITIONAL VISUAL GENERALIZATION UNDER OBJECT AND LOAD VARIATIONS

For completeness, we report the visual generalization experiment for Section 4.2. Here, we perturb all three loco–manipulation tasks by modifying only the manipulated objects and their loads while keeping the scene layout fixed. Specifically, (i) in Bag Packing, one of the paper bags is replaced with another bag of different appearance, size, and weight; (ii) in Box Loading, the bags inside the box are replaced with plastic containers; and (iii) in Cart Pushing, the carton on the cart is replaced with 60 kg barbell plates. We evaluate WholeBodyVLA, GR00T w/ LMO, and OpenVLA-OFT w/ LMO under these shifted conditions. As summarized in Fig. 6, WholeBodyVLA maintains the highest success rates across all perturbed settings, demonstrating that our framework remains robust even when the manipulated objects and loads differ noticeably from those seen during training.

Table 6: **Effect of removing proprioceptive state on the three tasks.** Each task is decomposed into two subgoals and an averaged score. "WholeBodyVLA w/o state" denotes a variant where the robot state is not injected into the action decoder.

| Method | Bag Packing | | Box Loading | | Cart Pushing | | Avg. Score |
|---|---|---|---|---|---|---|---|
| | Grasp Bags | Move & Squat | Squat & Grasp | Rise & Turn | Grab Handle | Push Ahead | |
| WholeBodyVLA w/o state | 21/25 | 12/25 | 22/25 | 14/25 | 24/25 | 22/25 | 76.7% |
| WholeBodyVLA | 23/25 | 13/25 | 19/25 | 17/25 | 23/25 | 22/25 | 78.0% |

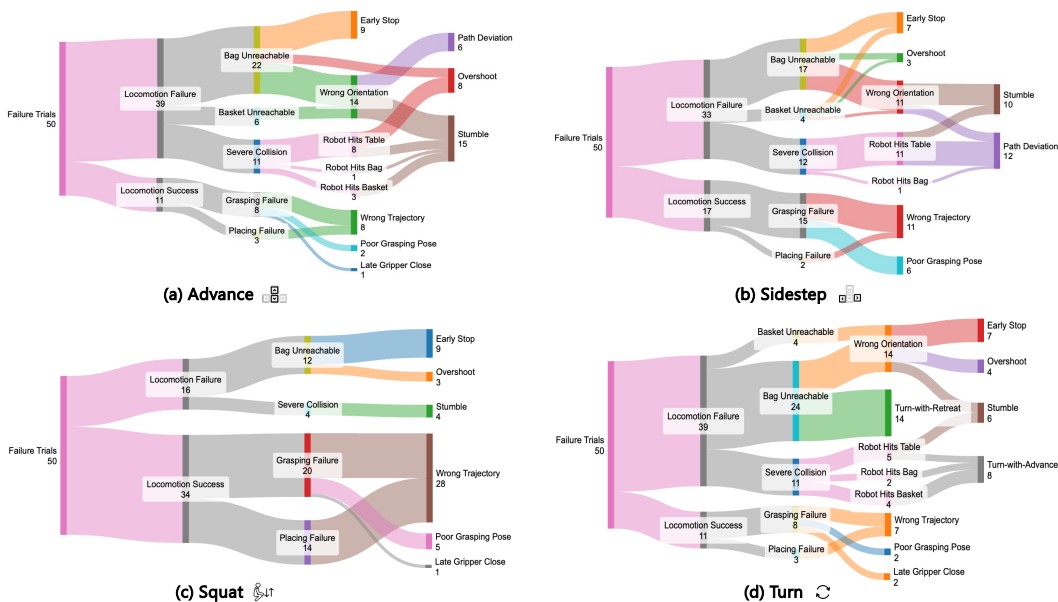

Figure 7: **Failure modes of WholeBodyVLA in start-pose generalization.** For each locomotion primitive—(a) advance, (b) sidestep, (c) squat, and (d) turn—we collect 50 failed trials from the start-pose generalization experiments (pick object and place into basket). Each Sankey diagram decomposes failures into locomotion vs. pick/place errors and finer-grained causes such as object/basket unreachable, wrong orientation, early stop, overshoot, collisions, stumble, and inaccurate grasp or placement.

Table 7: **Effect of removing proprioceptive state under visual variations.** Each task with visual variation (marked by *) is decomposed into two subgoals and an averaged score, using the perturbations described in Fig. 6.

| Method | Bag Packing * | | Box Loading * | | Cart Pushing * | | Avg. Score |
|---|---|---|---|---|---|---|---|
| | Grasp Bags | Move & Squat | Squat & Grasp | Rise & Turn | Grab Handle | Push Ahead | |
| WholeBodyVLA w/o state | 12/25 | 5/25 | 14/25 | 12/25 | 21/25 | 21/25 | 76.7% |
| WholeBodyVLA | 15/25 | 9/25 | 15/25 | 15/25 | 22/25 | 20/25 | 64.0% |

To further probe whether WholeBodyVLA truly relies on visual observations rather than the injected robot state, we compare the full model with a variant that does not feed the proprioceptive state into the action decoder, both in the original setting (Table 6) and under the visual variations of Fig. 6 (Table 7). Removing the state input increases variance and slightly degrades performance, especially in the visually perturbed conditions, but the w/o-state variant still achieves comparable task completion rates. This suggests that WholeBodyVLA has indeed learned to accomplish loco–manipulation tasks primarily from visual input, and that the visual generalization observed in Section 4.5 does not hinge on access to low-level proprioceptive information.

## C.3 FAILURE MODE ANALYSIS

To gain insight into the remaining limitations of WholeBodyVLA, we conduct a post-hoc failure analysis using the start-pose generalization experiments in Section 4.5. In this setting, the robot begins from diverse initial poses and must execute one approach primitive (advance, sidestep, squat, or turn) before picking an object from the table and placing it into a basket. For each primitive we gather 50 failed rollouts and manually annotate the underlying causes.

As illustrated in Fig. 7, failures are first grouped into *locomotion failures*—where the robot ends in an unsuitable stance for picking or placing—and *pick/place failures* that occur despite a reasonable stance. Each is further decomposed into more fine-grained categories such as *object unreachable*

Table 8: **Average execution duration (s) on the three tasks.**

| Method | Bag Packing | | Box Loading | | Cart Pushing | |
|---|---|---|---|---|---|---|
| | Grasp Bags | Move & Squat | Squat & Grasp | Rise & Turn | Grab Handle | Push Ahead |
| Modular Design | 19.2 | **23.0** | 21.5 | 7.9 | 12.0 | **11.7** |
| GR00T w/ LMO | 26.3 | 38.6 | 21.1 | 8.0 | 19.5 | 13.8 |
| OpenVLA-OFT w/ LMO | 23.6 | 35.9 | 33.2 | 13.8 | 16.9 | 13.1 |
| **WholeBodyVLA (ours)** | **18.4** | 29.7 | **16.8** | **7.6** | **11.3** | 12.7 |

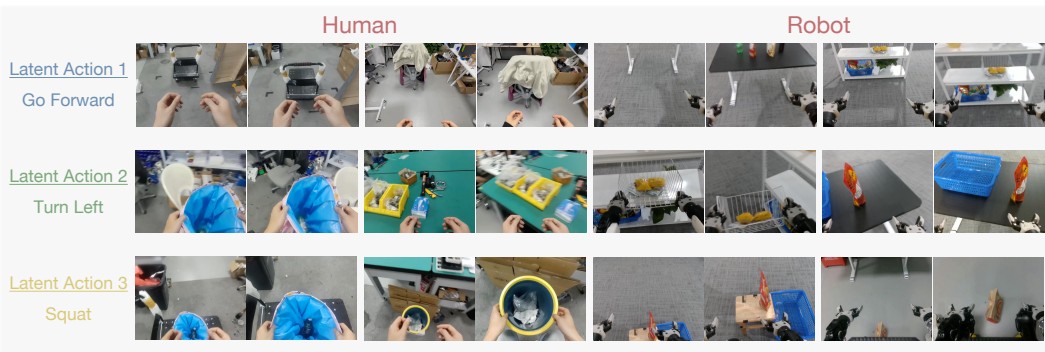

Figure 8: **Cross-domain retrieval with shared latent actions.** Human and robot clips retrieved for the same latent action (e.g., go forward, turn left, squat) exhibit consistent semantics across domains.

(often caused by early stop, path deviation, overshoot, or wrong orientation), *basket unreachable*, *severe collision* with the table or basket, *stumble*, or errors originating from the manipulation stage including *wrong reaching trajectory*, *poor grasp pose*, and *misaligned placement*.

Across the horizontal locomotion primitives (advance, sidestep, and turn), locomotion-related issues account for the majority of failures. Most unsuccessful episodes pass through the "object/basket unreachable" node, indicating that moderate stance or orientation deviations propagate into infeasible pick-and-place attempts. Severe collisions or stumbles occur much less frequently. In the squat primitive, failures are more evenly divided between locomotion (primarily incorrect final height or contact during descent) and pick/place errors caused by imprecise arm trajectories or grasp alignment.

Although the analysis is conducted within the start-pose generalization setup, the resulting patterns reflect broader limitations of the current model: the dominant failure modes arise from small but systematic stance and orientation errors during approach, rather than catastrophic behaviors. Improving approach precision—especially for turning, lateral stepping, and squatting—is expected to directly reduce downstream manipulation failures in future versions of the system.

### C.4    TASK EXECUTION TIME

Besides success rates, we also report execution time, averaged over the five successful trials for each subgoal. For every successful trial, we measure the wall-clock duration from the moment the instruction is issued to the moment the subgoal is judged successful. Table 8 summarizes the average execution time aggregated over all subgoals within each task.

### C.5    SHARED LATENT ACTION SPACE ACROSS EMBODIMENTS

Our three-stage training pipeline naturally leads to a common latent action space for human and robot data. Stage I trains VQ-VAE LAMs as inverse dynamics models from purely visual videos: the encoder consumes $(x_t, x_{t+k})$ pairs and the discrete code summarizes the visual change between frames. Stage II reuses the same videos and LAMs to supervise the VLA, which learns to predict these codes from images and language. Stage III then introduces Agibot X2 teleoperation data to

associate latents with robot joint targets and locomotion commands. Because the codes depend only on frame-to-frame visual motion rather than embodiment-specific joint values, the same latent can align human and robot motions. Figure 8 illustrates this behavior. For the latent action "Go Forward", the retrieved clips consistently include semantically aligned segments across both human-collected videos (left) and teleoperated robot demonstrations (right). Despite coming from different sources, these clips all correspond to the same underlying latent action. Together with the strong gap between WholeBodyVLA and "w/o LAM" in Table 2, this supports that the learned latent space is shared and transferable.

## C.6 SHARED LATENT ACTION SPACE ACROSS EMBODIMENTS

Apart from directly evaluating the downstream performance of the VLA, we introduce a convenient and efficient metric, the *Relative Reconstruction Gain* (RRG), to directly assess the quality of the LAM. Formally, we measure the reconstruction error of the predicted future frame $\hat{o}_{t+k}$ against the ground-truth $o_{t+k}$ as $\mathrm{MSE}_{\mathrm{reco}} = \mathrm{MSE}(\hat{o}_{t+k}, o_{t+k})$. For comparison, we define a simple temporal baseline that estimates the later frame by directly copying the former, with error $\mathrm{MSE}_{\mathrm{base}} = \mathrm{MSE}(o_t, o_{t+k})$. The proposed RRG is then given by:

$$\mathrm{RRG} = \frac{\mathrm{MSE}_{\mathrm{base}} - \mathrm{MSE}_{\mathrm{recon}}}{\mathrm{MSE}_{\mathrm{base}}}, \tag{6}$$

which captures the relative error reduction achieved by the LAM over the temporal baseline. A higher RRG indicates that the LAM provides more predictive latent codes, serving as an efficient proxy for assessing LAM quality without relying solely on downstream policy evaluation.

We evaluate RRG on downstream task videos, as shown in Table 9. Our recipe outperforms the shared one on both manipulation and locomotion splits, since the shared LAM underperforms the separate models on both manipulation and locomotion primitives. We consider this suggests inherent conflicts between locomotion-specific and manipulation-specific objectives when trained jointly.

Table 9: **RRG (%) across three tasks.** Each task is decomposed into primitives. Our separately trained LAMs show better downstream reconstruction performance than the shared alternative.

| Method | Bag Packing | | | | Box Loading | | | | | Cart Pushing | |
|---|---|---|---|---|---|---|---|---|---|---|---|
| | Grasp | Move | Squat | Place | Squat | Grasp | Rise | Turn | Place | Grab | Push |
| manip. lam | **21.78** | 23.64 | 18.71 | 24.73 | 21.22 | **25.09** | 22.92 | 28.15 | 23.96 | **19.12** | 19.92 |
| shared lam | 19.70 | 23.58 | 20.62 | **25.69** | 19.41 | 17.38 | 23.43 | 27.68 | 23.61 | 18.49 | 17.79 |
| **loco. lam** | 16.39 | **25.77** | **29.46** | 20.60 | **22.72** | 20.24 | **25.40** | **30.74** | **24.81** | 15.27 | **20.27** |

