# OpenReview forum: "WholeBodyVLA: Towards Unified Latent VLA for Whole-body Loco-manipulation Control"
_ICLR.cc/2026/Conference — ICLR 2026 Poster_

### Official Review · Reviewer_1asV · 2025-10-26

**Soundness:** 3
**Presentation:** 2
**Contribution:** 2
**Rating:** 4
**Confidence:** 4

**Summary:**

This paper presents WholeBodyVLA, a unified framework for humanoid loco-manipulation. The framework consists of two main components: Unified Latent Learning and the Loco-Manipulation-Oriented (LMO) RL Policy. In Unified Latent Learning, two separate VQ-VAE–based latent models are trained on manipulation and locomotion videos. The LMO policy is a reinforcement learning controller that executes discrete locomotion commands and adopts a two-stage curriculum to enhance training stability and precision. During inference, the VLA model decodes latent actions into upper-body joint angles and locomotion commands, which the LMO policy translates into torques for the robot’s lower body. Experiments conducted on the AgiBot X2 humanoid across three real-world loco-manipulation tasks demonstrate that WholeBodyVLA outperforms both modular and end-to-end baselines.

**Strengths:**

- The paper addresses an interesting and timely topic in humanoid robot control, focusing on whole-body loco-manipulation, which is of high relevance to the embodied AI and robotics communities.

- The proposed method is conceptually straightforward and well structured, making it relatively easy to reproduce and extend in future research.

- The experimental results demonstrate that the proposed method outperforms several strong baselines, indicating the effectiveness and practicality of the approach.

- The paper includes comprehensive ablation experiments that provide necessary insights into the contribution of each component.

**Weaknesses:**

The paper could benefit from clearer elaboration and justification of certain key arguments:
- The authors identify data scarcity and instability in RL controllers as the two main challenges in humanoid loco-manipulation. However, the connection between data scarcity and degraded loco-manipulation performance is not sufficiently explained or empirically verified. It would be helpful to clarify how the lack of data leads to poor generalization and to provide stronger evidence that the proposed use of video-based learning effectively mitigates this limitation.
- Some sections of the method description could be written more clearly, particularly regarding model training details and implementation specifics (see “Questions” section for further comments).

The locomotion component is relatively simple, relying on a small set of discrete motion primitives. This limits the expressiveness and adaptability of the policy for more complex movements.

The experimental evaluation could be improved by including longer-horizon tasks that require extended locomotion or sequential coordination, which would better demonstrate the robustness and scalability of the proposed framework.

**Questions:**

- Is my understanding correct that the LAMs are trained using human egocentric videos, while the VLA (or VLM) is trained using teleoperation data? If so, could the authors clarify why and how human and robot data can share the same latent action space?

- Would it be possible to pretrain the LAM using the teleoperation dataset’s camera videos instead of (or in addition to) human egocentric data? This might provide a more consistent distribution between perception and control domains.

- Regarding the data scarcity issue, is the problem primarily addressed by pretraining the LAM on large-scale human video data to provide latent supervision for robot policy learning? Please confirm if this interpretation is correct.

---

> ### Author Response · Authors · 2025-11-26
> **Authors' Response to Reviewer 1asV**
>
> We appreciate your careful review and constructive feedback. The
> submission PDF has been updated, with all changes marked in blue. We respond to
> each of your concerns in detail below.
>
> > *${\color{BrickRed}\text{Weakness 1-1:}}$*  Connection between data scarcity and degrated loco–manipulation performance.
>
>  In Table 2, removing video-based latent learning drops average success from 78.0% (*WholeBodyVLA*) to 39.3% (“*w/o LAM*”) under the same teleoperation data and LMO controller, indicating that a policy trained only on scarce robot data generalizes poorly. In addition, the new scaling curves in Fig. 3 (Sec. 4.3) show that, for any fixed amount of finetuning real-robot teleoperation data, models with stronger latent pretraining on action-free videos consistently achieve higher success and can match or surpass no-LAM models that use substantially more teleop data, providing direct evidence that video-based latent learning alleviates data scarcity for loco–manipulation.
>
> > *${\color{BrickRed}\text{Weakness 1-2:}}$* Clarity regarding training details and implementation specifics.
>
> Thanks for the comment. We have revised the method description to clarify the training and implementation details. We also refer to our responses to specific questions below.
>
> In particular, we now (i) spell out the three-stage training pipeline and how baselines/ablations are configured in Sec. 4.1 (*Baselines* and *Training protocol*). Additional implementation can be found in App. B.
>
> > *${\color{BrickRed}\text{Weakness 2:}}$*  Locomotion component is relatively simple. This limits the expressiveness and adaptability of the policy for more complex movements.
>
> Agreed. Our objective is not to create the most expressive whole-body controller to achieve motions like dancing and kung-fu, but to design a **simple, robust, and VLA-friendly interface** for bipedal loco–manipulation, as described in Lines 91-106 and Lines 286–290. The LMO policy follows the standard paradigm widely used in humanoid loco–manipulation systems (e.g., HOMIE [1], AMO [2], FALCON [3], $R^2S^2$ [4]), which all track a small set of command like velocities and stance heights and outputs full lower-body actions at 50 Hz. Such a compact interface greatly simplifies high-level decision making while still yielding rich, continuous whole-body motion.
>
>
>
> Despite this simplicity, the controller is non-trivial and sufficiently expressive for our setting. Table 4 shows accurate tracking for any locomotion command, and ablations confirm that removing key design choices significantly degrades accuracy and stability. In real-world experiments, the same primitive set supports the new evaluations we added in the revision, including traversing uneven terrain made of soft foam, artificial grass, wooden planks, and loose gravel, as well as more complex loco–manipulation tasks such as long-horizon, closing a drawer while squatting, and retrieving objects from a shelf. New videos are provided at `https://wholebodyvla.github.io/#terrain-gen` [**[click here](https://wholebodyvla.github.io/#terrain-gen)**] and below.
>
> [1] Ben, Q., et al. HOMIE: Humanoid loco-manipulation with isomorphic exoskeleton cockpit. RSS 2025.
> [2] Li, J., et al. AMO: Adaptive Motion Optimization for Hyper-Dexterous Humanoid Whole-Body Control. RSS 2025.
> [3] Zhang, Y., et al. FALCON: Learning Force-Adaptive Humanoid Loco-Manipulation. arXiv:2505.06776.
> [4] Zhang, Z., et al. Unleashing Humanoid Reaching Potential via Real-world-Ready Skill Space. arXiv:2505.10918.
>
> > *${\color{BrickRed}\text{Weakness 3:}}$* Evaluations could be improved by including longer-horizon tasks.
>
> Thank you for the suggestion. In the revision, we have added a long-horizon real-world task that explicitly stresses extended locomotion and sequential coordination, see video at `https://wholebodyvla.github.io/#long-horizon` [**[click here](https://wholebodyvla.github.io/#long-horizon)**]. The new task consists of **seven steps**: (1) go forward to the table; (2) grasp the snack bags; (3) turn left toward a storage drawer; (4) sidestep and walk to the front of the drawer; (5) place the snacks into the drawer; (6) squat down to brace both arms against the drawer front; and (7) push the drawer closed while maintaining balance. This sequence lasts around 1'50'' (compared to 50'' in initial tasks), spans a larger workspace, and includes coordinated whole body manipulation such as closing the heavy drawer from a squatting posture.

---

> ### Author Response · Authors · 2025-11-26
> **Authors' Response to Reviewer 1asV (continued)**
>
> > *${\color{BrickRed}\text{Question 1-1:}}$*  Are LAMs are trained using human videos, while the VLA trained using teleoperation data?
>
> Your understanding is mostly correct. To be precise, our training pipeline has three stages: (i) Stage I trains the LAMs purely on action-free human or web videos; (ii) Stage II uses these LAMs to annotate the same videos with latent actions and pretrains the VLA to predict these codes from images and language; (iii) Stage III then finetunes this latent-conditioned VLA on Agibot-X2 teleoperation data with true robot actions.
>
> > *${\color{BrickRed}\text{Question 1-2:}}$*  Why and how can human and robot data share the same latent action space?
>
> Human and robot data can share the same latent action space because the LAM encodes **frame-to-frame visual changes**, not embodiment-specific joint values. Prior work such as LAPA [1], UniVLA [2], and GR00T [3] shows that VQ-VAE models trained on heterogeneous videos learn compact codebooks whose codes correspond to abstract motion patterns (e.g., “go forward”) that are consistent across embodiments. Our LAM follows the same idea, using DINOv2 features and a small codebook encourages each latent to represent high-level primitives—walking forward, sidestepping, squatting, reaching—rather than low-level joint angles. These primitives naturally occur in both human egocentric footage and humanoid first-person views, enabling a shared latent action space.
>
> This can be evidenced in our experiments. We show that such latent supervision is effective, where WholeBodyVLA with LAM significantly outperforms the “w/o LAM” variant trained on exactly the same teleoperation data (78.0\% vs.  39.3\% average success in Table 2). In the revision, we additionally include cross-domain retrieval visualizations (Appendix C.5 Fig. 8), where a single locomotion latent retrieves semantically aligned motions (e.g., “go forward”, “turn”) across human egocentric videos and humanoid first-person observations.
>
> [1] Ye, S., et al. Latent Action Pretraining from Videos. ICLR 2025.
> [2] Bu, Q., et al. Learning to Act Anywhere with Task-centric Latent Actions. RSS 2025.
> [3] Bjorck, J., et al. GR00T N1: An Open Foundation Model for Generalist Humanoid Robots. arXiv:2503.14734.
>
> > *${\color{BrickRed}\text{Question 2:}}$*  Would it be possible to pretrain the LAM using the teleoperation dataset’s camera videos instead of (or in addition to) human egocentric data?
>
> In principle, it is possible to include teleoperation ego-camera videos when training the LAM. However, this would not address the core data-scarcity problem.
>
> 1. **Teleoperation data are intrinsically small-scale.** Each trajectory is expensive to collect, requiring a VR+joystick or mocap setup and a human operator with a robot, so the total number of such trajectories is typically only on the order of a few hundred (i.e., a few hours of data). By contrast, pretrained VLAs like GR00T and OpenVLA use more than 8,000 hours of diverse egocentric videos. To obtain a VLA with good generalization, pretraining must therefore rely on much larger and more diverse corpora, not just this small, robot-specific set. If the LAM were trained mainly on teleoperation videos, it would still be constrained by the same limited dataset as the policy.
> 2. **LAM is valuable because it unlocks large-scale action‑free videos.**  The main purpose of LAM is to turn abundant, cheap videos (egocentric human videos and open-source videos) into additional supervisory signals. When teleoperation videos already come with ground‑truth robot actions at essentially no extra cost, relabeling them with latent actions provides little extra benefit.
> 3. **Fair comparison with other pretrained VLAs.** VLA baselines (e.g., GR00T, OpenVLA-OFT) are pretrained on large-scale egocentric videos without any X2 teleoperation data. To keep the comparison fair, we follow the same protocol. WholeBodyVLA is also pretrained on action-free egocentric videos by our unified latent learning, and *only* Stage III introduces Agibot-X2 teleoperation data for all methods.
>
> > *${\color{BrickRed}\text{Question 3:}}$*  Is the data-scarcity issue primarily addressed by pretraining the LAM on large-scale human videos to provide latent supervision for robot policy learning?
>
> Yes. The main goal of unified latent learning is to mitigate the scarcity of expensive MoCap/teleoperation trajectories by first training LAMs on large-scale, action-free egocentric videos and then using their latent codes as supervision for VLA pretraining. As noted in the paper (Lines 127–130, 147-148), this lets the policy acquire loco–manipulation knowledge from abundant, cheap videos, and only the final finetuning stage relies on a small amount of Agibot-X2 teleoperation data to adapt the latent-conditioned policy to our robot and controller.

---

> > ### Comment · Reviewer_1asV · 2025-11-27
> >
> > Thank you for your response, now I have a better understanding of the paper.
> >
> > Here are some details that I am not clear:
> > 1. when using LAM for robot training, I think locomotion action can come from video since it should be root velocity, but where does the upper body action come from?
> >
> > 2. Are the locomotion and upper body share same decoder? Would it be helpful if you split upper body and lower body entirely, and use different encoder/decoders?
> >
> > 3. please correct me if I am wrong. The logic of this paper is that the teleoperation data is limited, so this work propose to use human egocentric manipulation-aware locomotion videos to help training VLA. And the work build a framework to show how these videos should be used to help the training.

---

> > > ### Author Response · Authors · 2025-11-27
> > > **Thanks for your prompt discussion**
> > >
> > > Thanks for your follow-up questions. We clarify them below.
> > >
> > > > *${\color{BrickRed}\text{Question 1:}}$*  Where does the upper body action come from?
> > >
> > > We train manipulation LAM on the large-scale robotic dataset AgiBot World, motivated by prior manipulation works (e.g. LAPA [1], UniVLA [2], and GR00T [3]). The LAM captures the motion of the robot arm, end-effector, and manipulated objects in the video, and encodes these dynamics into latent actions. Both the manipulation LAM and locomotion LAM then supervise the VLA backone. Additional discussion and details can be found in the paper (Lines 242–254 and 889–890).
> > >
> > >
> > > [1] Ye, S., et al. Latent Action Pretraining from Videos. ICLR 2025.
> > > [2] Bu, Q., et al. Learning to Act Anywhere with Task-centric Latent Actions. RSS 2025.
> > > [3] Bjorck, J., et al. GR00T N1: An Open Foundation Model for Generalist Humanoid Robots. arXiv:2503.14734.
> > >
> > > > *${\color{BrickRed}\text{Question 2:}}$*  Do locomotion and upper-body actions share the same decoder? Would splitting help?
> > >
> > > In this question, we understand the reviewer as referring to the **VLA action decoder**, rather than the LAMs. The LAMs for locomotion and manipulation are already trained separately, but the final **VLA action decoder is shared**, outputting both upper-body joint actions and the locomotion command.
> > >
> > > We intentionally keep this action decoder unified.
> > > - If use a single VLA backbone but split only the final action decoder into two decoders (upper vs. lower body), this would mainly increase parameters without changing the underlying representation, while introducing synchronization and consistency issues between two heads that must describe one whole-body action.
> > > - If instead split the VLA itself into two independent models (upper vs. lower body), the system would no longer be truly unified: each policy could issue incompatible intentions (e.g., initiating a step while the arms are still engaged in manipulation). Two separate VLAs inherently complicate temporal consistency and coherent whole-body control.
> > >
> > > For these reasons, we keep a shared VLA and action decoder to preserve whole-body coordination.
> > >
> > > > *${\color{BrickRed}\text{Question 3:}}$*  Please correct me if I am wrong. The logic of this paper is that the teleoperation data is limited, so this work propose to use human egocentric manipulation-aware locomotion videos to help training VLA. And the work build a framework to show how these videos should be used to help the training.
> > >
> > > Yes. We use large-scale action-free videos to compensate for the limited teleoperation data and to guide VLA pretraining. These videos cover **both** open-source manipulation video and human manipulation-aware locomotion video, so the learned latents provide priors for arms and legs jointly. After finetuning on teleop data, the resulting VLA is used to solve integrated loco–manipulation tasks.

---

> ### Comment · Reviewer_1asV · 2025-11-27
>
> Thanks for the reply.
>
> I would conditional adjust my score if you could reformulate the story of the paper.
>
> Here is my thought, although the  data scarcity and unstable RL training is relevant to your work, they are not the main problem you addressing. And these two limitation you raise in the abstract and introduce does not directly reflect in the experiment section. Line340-345 seem to have no correspond to the current limitations you claim. In summary, I feel the story is not coherent and somehow I miss the middle.
>
> Maybe you can structure the paper as 'how you use egocentric videos for training VLA'. Therefore the flow is:
> Teleoperation data is limited -> use egocentric video -> you have your framework that use the unified latent + RL structure -> you show your framework is effective.
>
> Not sure if I made my point clear.

---

> > ### Author Response · Authors · 2025-12-03
> >
> > Thank you for the suggestion. We have revised the paper with the key changes highlighted in blue to center the story on how egocentric videos are used to train a whole-body VLA, following the flow: on the data side, *teleoperation data scarcity → learn from low-cost action-free videos → latent learning*; on the control side, *unstable execution → VLA decision–controller execution mismatch → LMO controller*. We also refined the structure and presentation of the experiments and ablations to better align with this storyline.

---

### Official Review · Reviewer_76f3 · 2025-10-31

**Soundness:** 4
**Presentation:** 3
**Contribution:** 3
**Rating:** 8
**Confidence:** 3

**Summary:**

This paper introduces WholeBodyVLA, a unified Vision-Language-Action (VLA) framework that enables bipedal humanoid robots to perform large-space loco–manipulation tasks (e.g., squatting to grasp, turning to place, pushing 50 kg carts).
The method integrates two key innovations: 1. Unified Latent Learning (ULL) — jointly learns locomotion and manipulation latent actions from action-free egocentric videos. 2. Loco–Manipulation–Oriented (LMO) RL Policy — replaces conventional continuous velocity tracking with a discrete command interface and introduces structured upper-body perturbations from manipulation data to improve balance and stability. Experiments on the Agibot X2 humanoid show that WholeBodyVLA outperforms modular and end-to-end baselines (e.g., GR00T N1.5, OpenVLA-OFT) by 21.3–24 % in success rate, while maintaining robust multi-tasking and visual generalization under unseen conditions (e.g., different box contents or weights).

**Strengths:**

1.	The paper clearly identifies the missing link between manipulation-aware locomotion and VLA-based manipulation, proposing the first end-to-end unified framework that integrates both in real-world humanoid control.
2.	The discrete command interface + structured perturbation RL is elegantly engineered. The two-stage curriculum and well-defined reward shaping (directional accuracy, stand-still penalty) demonstrate deep insight into locomotion stability and precision.
3.	Evaluation on multiple real-world tasks (bag packing, box loading, cart pushing) and ablations (without RL, with velocity-based RL, with shared LAM, etc.) show both component-wise contributions and robustness. The use of both real-robot and MuJoCo simulations strengthens credibility.

**Weaknesses:**

1.	The upper-body movements demonstrated in the three tasks appear quite limited — the shoulder seems to move primarily along the pitch axis. However, since the tabletop objects can be positioned in more diverse locations, introducing richer interaction motions could better showcase the capabilities of the VLA.
2.	The current tabletop task is rather simple and singular. Did the authors also collect additional related data that could be used to supplement or expand the current manipulation tasks?

**Questions:**

See Weakness

---

> ### Author Response · Authors · 2025-11-26
> **Authors' Response to Reviewer 76f3**
>
> Thank you for taking the time to provide helpful comments. We have added new experiments in the paper and highlighted all modifications in blue in the updated
> PDF. Our responses to your points are given below.
>
> >  *${\color{BrickRed}Weakness 1:}$* Objects can be positioned in more diverse locations.
>
> We have added new experiments on position generalization, involving grasping objects placed at different positions (see videos at `https://wholebodyvla.github.io/#obj-gen` [**[click here](https://wholebodyvla.github.io/#obj-gen)**]) as well as additional wiping coffee stains located at different positions on the tabletop (`https://wholebodyvla.github.io/#whats-more` [**[click here](https://wholebodyvla.github.io/#whats-more)**]).
> Moreover, the quantitative result of  in Fig. 3 consistently show the advantage of WholeBodyVLA over the other baselines.
>
> > *${\color{BrickRed}Weakness 2:}$* The current tabletop task is rather simple and singular.
>
> Thank you for the suggestion. In the revision we added several more complex everyday loco–manipulation tasks (`https://wholebodyvla.github.io/#whats-more` [**[click here](https://wholebodyvla.github.io/#whats-more)**]), including wiping stains on a table, vacuum cleaning, watering plants, and retrieving objects from a shelf, as well as a long-horizon task (`https://wholebodyvla.github.io/#long-horizon` [**[click here](https://wholebodyvla.github.io/#long-horizon)**]) that ends with closing a heavy drawer while squatting (coordinated whole-body manipulation). These new tasks and their success rates are summarized in Sec. 4.5 (bottom row of Fig. 3).

---

### Official Review · Reviewer_N1iB · 2025-11-01

**Soundness:** 4
**Presentation:** 4
**Contribution:** 2
**Rating:** 4
**Confidence:** 4

**Summary:**

The paper proposes WholeBodyVLA, a unified VLA framework for end-to-end humanoid loco-manipulation on Agibot X2. To address data scarcity for manipulation-aware locomotion, the authors introduce unified latent learning with two VQ-VAE latent action models (separately for manipulation and locomotion). For precise and stable execution, a Loco–Manipulation–Oriented (LMO) RL controller with a discrete command interface is trained as the lower-body controller. Real-robot experiments on three suites (bag packing, box loading, cart pushing) are reported, where WholeBodyVLA achieves higher success rates than modular and open-source end-to-end baselines.

**Strengths:**

1. Clear paper writing and problem formulation. The paper identifies the gap between modular pipelines and true end-to-end manipulation-aware locomotion, motivating why coupling is necessary for stability and task success.2. Unified latent learning with modality-separated LAMs. The paper identified the sub-optimality of using mixed-data to train a single LAM and propose to train separate LAMs for manip and locomotion
2. Separation of latent action spaces. Training distinct latent action models for manipulation/locomotion is a sensible design choice that addresses mixed-motion dataset and improves learning for the high-level policy.
3. Lower-body controller tailored for loco-manipulation. The LMO RL controller with a discrete command interface is well aligned with whole-body objectives (heading, position, stance), and the design is empirically validated.
4. Real-robot evaluation. Experiments on physical hardware across multiple task families strengthen external validity and demonstrate the practicality of the method.

**Weaknesses:**

1. Ambiguity in loco-manipulation demands of the tasks. In all three demos, target objects appear within immediate arm’s reach at the start, so grasping could be completed without meaningful locomotion. Even though the side stepping motion is stable, it is still difficult to determine whether improvements arise from integrated loco-manipulation versus decoupled manipulation with stepping.
2. Baseline fairness. Some baselines were not evidently fine-tuned on the same Agibot/LMO data or adapted to the same low-level interface. Since GR00T was originally trained with the data collected by NVIDIA's own low-level controller, directly plugging into LMO can be suboptimal or OOD. A stronger comparison would fine-tune GR00T/OpenVLA on the same teleop/robot dataset (of AgiBot) and interface, with matched training distribution.
3. Limited analysis of latent-learning/data scaling. The paper motivates latent action learning as a solution for data scarcity, but there is no study of dataset size (with/without action labels) versus performance, nor a comparison of scaling first-person video versus teleop data. This limits conclusions about where latent learning delivers the biggest gains.

**Questions:**

1. Could you include real-world trials with ≥1 m initial separation (for example, robot stand 1m away from the table/cart) for all demos? Plots of success vs. start distance would help.
2. For GR00T/OpenVLA baselines, can you fine-tune on the identical Agibot/LMO data? If already done, please add these details.
3. Can you provide scaling curves for (a) total hours of action-free video for latent learning and (b) hours of robot/teleop data, and ablate the contribution of each?
4. The current low-level LMO policy appears tuned for flat-ground walking and squatting motions. How does the approach generalize to more complex whole-body coordination tasks such as stair climbing, uneven terrain traversal, or heavy object lifting (which is common for loco-manipulation) that require significant upper–lower body coupling? If such generalization is not yet supported, could the authors discuss what architectural or training changes would be necessary to extend the controller to these regimes?

---

> ### Author Response · Authors · 2025-11-26
> **Authors' Response to Reviewer N1iB**
>
> Thank you for your thoughtful and detailed review. We have updated the submission PDF and highlighted all revisions in blue. We address your comments point by point below.
>
> > *${\color{BrickRed}\text{Weakness 1:}}$*  Ambiguity in loco-manipulation demands.
> > *${\color{BrickRed}\text{Question 1:}}$*   ≥1m initial-distance trials and success-vs-distance plots.
>
> Thanks for the question. We have added new structured real-world trials in Sec. 4.3 where the robot starts from a range of start-poses that vary not only the X-axis distance to the table, but also the Y-axis offset, yaw angle, and table height. Please also see videos at `https://wholebodyvla.github.io/#dist-gen` [**[click here](https://wholebodyvla.github.io/#dist-gen)**]. We provide detailed results below (Table R1).
>
> Table R1: Success rate (%) vs. start distance (X-axis) on the approach-pick-place task.
>
> | Method               | 1 m | 1.25 m | 1.5 m |
> |----------------------|:-----:|:-----:|:-----:|
> | Modular Design       |   68  |  60   |  56   |
> | WholeBodyVLA w/o LAM |  52   |  40   |  36  |
> | **WholeBodyVLA**     | **80**| **80**| **72**|
>
> WholeBodyVLA maintains high success with only modest degradation as the start distance increases to and beyond 1m, and significantly outperforms decoupled manipulation with stepping (*Modular Design*). We also note that the original settings requires integrated loco-manipuation and are non-trivial to succeed, as demonstrated by the degenerated performance of baselines in Table 2. Conceptually, the robot is controlled entirely by a single VLA that must autonomously infer from first-person images and language **when to walk, how far to move, how to align its heading, and when to stop** which fully determines whether the robot ends in a good **in-domain configuration for picking or placing**.
>
> > *${\color{BrickRed}\text{Weakness 2:}}$*  Baseline fairness.
> > *${\color{BrickRed}\text{Question 2:}}$*  Fine-tune GR00T/OpenVLA on identical Agibot/LMO data.
>
> The comparison in Table 2 is **fair**. All VLA baselines are fine-tuned on the same Agibot-X2 teleoperation data and adapted to the same LMO-based low-level interface. We have clarified the protocols in Sec. 4.1 (Training protocol).
>
> In more details,
> 1. **Pre-training stage: no Agibot-X2/LMO exposure**. All three models (GR00T, OpenVLA-OFT, or WholeBodyVLA) start from comparable generalist visuomotor priors, and none of them has seen any Agibot-X2 teleoperation data during pre-training.
>
> 2. **Fine-tuning stage: identical Agibot-X2/LMO setting**. For fairness, all baselines in Table 2 (including GR00T and OpenVLA) are fine-tuned on the *same* Agibot-X2 real-world dataset collected by teleoperation using the same LMO controller.
>
> This follows the standard usage of foundation VLAs, where deployment always requires finetuning the action decoder on data from the target robot and its controller. The action decoder, which outputs upper-body joint actions and the locomotion command, is only optimized in this finetuning stage for all methods. Moreover, the data you mentioned collected by NVIDIA's own low-level controller for GR00T was only fixed-base tabletop videos for pretraining. The controller is used merely to teleoperate arms and does not involve whole-body locomotion control.
>
> Under matched setting, WholeBodyVLA substantially outperforms both GR00T/OpenVLA and its ablated baselines, indicating that the gains come from the proposed latent learning.
>
> > *${\color{BrickRed}\text{Weakness 3}}$* & *$\color{BrickRed}\text{Question 3:}$*  Limited analysis of latent-learning/data scaling.
>
> Thank you for the suggestion. In the revision we add a scaling study in the new Fig. 3 (Sec. 4.3). The top plots report average success rate under start-pose and appearance/layout shifts, where the horizontal axis is the amount of teleoperation data used for finetuning, and each curve corresponds to a different amount of LAM pretraining data (0\%, 25\%, 50\%, 100\% of the locomotion or manipulation latent). Details are given in App. C.1.
>
> The curves show a clear pattern: for any fixed amount of finetuning data, stronger latent pretraining consistently yields higher success, and models with full LAM pretraining can match or exceed the performance of no-LAM models that use substantially more teleop data. This provides direct evidence that latent action learning leverages large action-free video to reduce the amount of robot data needed for good loco–manipulation generalization.

---

> ### Author Response · Authors · 2025-11-26
> **Authors' Response to Reviewer N1iB (continued)**
>
> > *${\color{BrickRed}\text{Question 4:}}$*  Generalization to difficult terrains and tasks.
>
> Thanks. We have added new experiments involving traversing four different terrain types (`https://wholebodyvla.github.io/#terrain-gen` [**[click here](https://wholebodyvla.github.io/#terrain-gen)**]), closing a heavy drawer while squating (`https://wholebodyvla.github.io/#long-horizon` [**[click here](https://wholebodyvla.github.io/#long-horizon)**]) and other more complex tasks (`https://wholebodyvla.github.io/#whats-more` [**[click here](https://wholebodyvla.github.io/#whats-more)**]) in Sec. 4.5.
>
> Notably, even in seemingly simple tasks (e.g., Bag Packing), the robot performs non-trivial whole-body coordination: the success of the final placing step depends critically on the accuracy of the preceding locomotion—approach distance, heading alignment, and squat depth—all inferred end-to-end by a single VLA policy. These behaviors are exactly the upper–lower-body coupling our method targets.
>
> For tasks like stair climing and heavy object lifting, they require terrain-specific RL training [1] or rigid task-specific end-effector designs [2]. Our framework can incorporate such capabilities by extending LMO training or hardware improvements in future works.
>
> [1] Long, J., et al. Learning Humanoid Locomotion with Perceptive Internal Model. ICRA 2025.
> [2] Zhang, Y., et al. FALCON: Learning Force-Adaptive Humanoid Loco-Manipulation. arXiv:2505.06776.

---

### Official Review · Reviewer_C1fR · 2025-11-01

**Soundness:** 2
**Presentation:** 3
**Contribution:** 2
**Rating:** 2
**Confidence:** 4

**Summary:**

WholeBodyVLA addresses the challenge of enabling humanoid robots to perform large-space whole-body loco–manipulation, which requires tight integration of precise locomotion and dexterous manipulation. The authors identify two key limitations in prior work: (1) the scarcity of data that jointly captures manipulation-aware locomotion, and (2) the instability and imprecision of existing RL controllers due to continuous velocity-tracking objectives and unstructured upper-body perturbations. To overcome these, they propose a unified latent learning framework that trains separate Latent Action Models (LAMs) for manipulation and locomotion on low-cost, first-person videos, thereby enabling a vision-language-action (VLA) policy to jointly predict upper- and lower-body commands. For stable execution, they introduce a Loco–Manipulation–Oriented (LMO) RL policy featuring a discrete command interface (e.g., forward/turn/squat) and structured perturbations derived from real manipulation trajectories. Evaluated on the AgiBot X2 humanoid, WholeBodyVLA autonomously completes multi-step loco–manipulation tasks, including squatting to grasp a box, placing it onto a cart, and pushing a load exceeding 50 kg. The method outperforms strong end-to-end and modular baselines by 21.3% on average and demonstrates robust generalization under unseen visual conditions.

**Strengths:**

**Originality**

WholeBodyVLA introduces a unified vision-language-action framework for humanoid loco-manipulation, extending latent action learning from tabletop manipulation to full-body coordination. The work’s key novelty lies in the dual-LAM design, which explicitly separates locomotion and manipulation latent spaces to mitigate conflicts between camera ego-motion and hand–object motion—an insightful adaptation that broadens latent learning’s applicability. Additionally, the proposed Loco-Manipulation-Oriented (LMO) reinforcement learning scheme combines a discrete command interface and structured upper-body perturbations to achieve precision, start–stop control, and directional stability, marking a tailored optimization for humanoid whole-body control. While the system’s components (VQ-VAE, DINOv2, RL control) are not inherently new, their orchestration for large-space humanoid manipulation represents a meaningful, though evolutionary, advance with the first systematic real-world validation.

**Quality**

Experiments cover three multi-step real-world tasks with ablations on control precision, stability, and generalization under visual perturbations. While comprehensive, the analysis lacks deeper investigation into sample efficiency, LAM hyperparameter sensitivity, or scalability beyond ~50 teleoperated demos per task. The results validate feasibility but not robustness or data efficiency, limiting the strength of empirical claims.

**Clarity**

The paper is well-structured and includes helpful runtime details (e.g., 10 Hz VLA, 50 Hz RL). However, critical parameters (codebook size, temporal stride k, reward weights) are buried in appendices, hindering reproducibility. Moreover, the interaction between locomotion and manipulation latents—central to the “unified” claim—remains underspecified, reducing interpretability.

**Significance**

The work addresses a key challenge in humanoid autonomy: manipulation-aware locomotion in a unified perception-to-control pipeline. Its real-world demonstration of multi-step, heavy-load tasks offers tangible engineering significance. While not a conceptual leap, it represents an important integration step toward practical deployment.

**Weaknesses:**

**Insufficient Evidence for “Unified” Whole-Body Control**

Despite the “unified” claim, the system decouples upper-body manipulation and lower-body locomotion: the VLA predicts separate manipulation and locomotion latents, a lightweight decoder outputs arm joint targets and a discrete locomotion command, and the LMO policy executes lower-body torques. This constitutes co-scheduled rather than end-to-end joint whole-body control, relying on downstream modules to handle cross-coupling.

**Limited Generalization and Robustness Evaluation**

Visual generalization is evaluated only under narrow appearance or load variations, such as unseen bags or cart weights. However, critical distribution shifts, including changes in scene layout, terrain friction, lighting, occlusion, or moving obstacles, are not tested, which limits the assessment of real-world robustness.

**Non-standardized Tasks & Metrics Hinder Comparability**

All three tasks (bag packing, box loading, cart pushing) are custom-designed with ad-hoc subgoal scoring, lacking standardized benchmarks or community-agreed metrics. This hinders reproducibility and cross-method comparison.

**No Failure Mode Analysis**

Table 2 shows numerous subgoal-level failures across phases, yet the paper provides no breakdown of failure modes such as approach, grasp, turn, squat, or push to identify dominant bottlenecks.

**Questions:**

**Questions**

**Q1: Failure Mode Distribution** – In the reported 25×6 subgoal evaluation, what proportion of failures arises from each stage (approach, grasp, turn, squat, push)? Can the authors provide a quantitative breakdown and correlate it with LMO directional and stance-height errors to reveal whether locomotion deviations systematically cause manipulation failures?

**Q2: Visual Generalization Boundary** – Beyond the tested appearance and load perturbations, has the system been evaluated under changes in ground friction, slope, lighting, occlusion, or dynamic obstacles? What are the success rates and degradation patterns under such environmental variations?

**Suggestions**

**S1: Detailed Failure Taxonomy** – Include a frequency breakdown of failures by task stage and relate them to quantitative LMO metrics (heading deviation, height tracking). This would help identify dominant failure patterns and clarify coupling between locomotion and manipulation.

**S2: Broader Robustness Evaluation** – Extend experiments to cover terrain/friction variation, slopes, lighting/occlusion, and moving obstacles, providing explicit success curves or degradation trends to substantiate robustness beyond appearance shifts.

**S3: Include standardized benchmarks and metrics** – Complement author-defined tasks with community-recognized scenarios and quantitative metrics (e.g., path/pose accuracy, time-to-goal, safety violations), enabling reproducible cross-paper comparison beyond the custom subgoal scores.

---

> ### Author Response · Authors · 2025-11-26
> **Authors' Response to Reviewer C1fR**
>
> Thank you for your detailed review. We have updated the
> submission PDF and highlighted all revisions in blue. We address your comments
> point by point below.
>
> > *${\color{BrickRed}\text{Weakness 1:}}$* Insufficient Evidence for “Unified” Whole-Body Control.
>
> We clarify that our claim is **unified latent VLA** throughout the submission, not “unified whole-body control.”  In our statement, unified refers to **unified latent-level decision-making for both manipulation and locomotion**.
>
> We model the two modalities jointly through
>
> $\pi_\theta(c^{\text{mani}}_t, c^{\text{loco}}_t \mid o_t, \ell)$,  in **Line 260**,
>
> predicting both latents in one forward pass (Sec. 3.1; Fig. 2). An action decoder uses both latents and the robot state to output arm joint targets and a discrete locomotion command.
>
> Our LMO RL controller does not replace this coupling — it simply ensures the feasibility and stability of the locomotion latent chosen by the unified VLA. Removing this unified latent design causes large degradation (e.g., w/o LAM: **39.3%** vs. WholeBodyVLA: **78.0%**, Table 2), confirming its necessity. Currrent VLA work is either manipulation-only (RT-2 / OpenVLA / GR00T, mostly fixed-base arms) or navigation-only (VLN), and to our knowledge no prior model jointly decides arm manipulation actions and legged locomotion for humanoids.
>
> > *${\color{BrickRed}\text{Weakness 2:}}$*  Limited Generalization and Robustness Evaluation.
> > *${\color{BrickRed}\text{Question 2}}$* & *${\color{BrickRed}\text{Suggestion 2:}}$*  Visual generalization boundary & broader robustness.
>
> We have added more standardized generalization experiments in Sec. 4.3 during rebuttal. Please see new videos at `https://wholebodyvla.github.io/#generalization` [**[click here](https://wholebodyvla.github.io/#generalization)**]. We add generalization evaluations under three aspects.
>
> 1. Systematic **start-pose shifts**: X-axis/Y-axis offsets, yaw angles, and table heights. The robot must compose advance, sidestep, turn, and squat to reach a good manipulation configuration.
> 2. **Background and layout**: We add tabletop distractors, change table appearance, and introduce unseen objects with different colors, shapes, and categories at randomized poses.
> 3. **Environment and contact variations**: Moving pedestrians, walking around fixed occluders (`https://wholebodyvla.github.io/#navigation` [**[click here](https://wholebodyvla.github.io/#navigation)**]), traversing uneven terrain made of foam, artificial grass, wooden planks, and loose gravel (`https://wholebodyvla.github.io/#terrain-gen` [**[click here](https://wholebodyvla.github.io/#terrain-gen)**]), and carrying out more complex everyday tasks (`https://wholebodyvla.github.io/#whats-more` [**[click here](https://wholebodyvla.github.io/#whats-more)**])  such as wiping a table or mopping a marked area.
>
> Across these scenarios, WholeBodyVLA maintains high average success rate with only modest degradation (new Fig. 3 in Sec. 4.3).
>
> > *${\color{BrickRed}\text{Weakness 3:}}$*  Non-standardized Tasks & Metrics Hinder Comparability.
> > *${\color{BrickRed}\text{Suggestion 3:}}$*  Need for community-recognized scenarios and quantitative metrics.
>
> We agree that standardized benchmarks would benefit fair comparisons and future humanoid loco–manipulation work. This has been a long-standing challenge for the robotics community, and there is currently no such benchmarks for humanoid loco–manipulation.  We would like to note that prior related studies (e.g., HOMIE [1], AMO [2], FALCON[3], $R^2S^2$ [4]) all rely on custom real-world setups.
>
> Regarding metrics, the primary quantitative measure used across embodied/humanoid real-robot research is *task success rate*, which we follow for comparability. For the additional metrics mentioned:
>
> - *Path/pose accuracy*: Due to the multi-modality of behaviors in real-world robot motion, there is **no canonical ground-truth trajectory**. Tasks like “walk over and pick up the box” admit **countless** valid paths and poses, making *path/pose accuracy* ill-defined. For LMO controller with definable references, we reported tracking-error metrics in simulation (Table 4).
> - *Time-to-goal*: We provide average episode duration for each task in Appendix C.4 (Table 9).
> - *Safety violations*: Prior humanoid loco–manipulation work does not report a standardized “safety-violation” metric. But we have added detailed failure-case analyses in Appendix C.3 (Fig. 7), which details the frequency and types of collisions or stumbles observed.
>
> [1] Ben, Q., et al. HOMIE: Humanoid loco-manipulation with isomorphic exoskeleton cockpit. RSS 2025.
> [2] Li, J., et al. AMO: Adaptive Motion Optimization for Hyper-Dexterous Humanoid Whole-Body Control. RSS 2025.
> [3] Zhang, Y., et al. FALCON: Learning Force-Adaptive Humanoid Loco-Manipulation. arXiv:2505.06776.
> [4] Zhang, Z., et al. Unleashing Humanoid Reaching Potential via Real-world-Ready Skill Space. arXiv:2505.10918.

---

> ### Author Response · Authors · 2025-11-26
> **Authors' Response to Reviewer C1fR (continued)**
>
> > *${\color{BrickRed}\text{Weakness 4:}}$*  No Failure Mode Analysis.
> > *${\color{BrickRed}\text{Question 1:}}$* & *${\color{BrickRed}\text{Suggestion 1:}}$*  Need for stage-wise failure frequencies and relation to LMO metrics.
>
> Thanks for the suggestion. We have added an explicit failure-mode analysis in Appendix C.3 (Fig. 7). Using the start-pose generalization setting of Sec. 4.3, we collect 50 failed rollouts for each approach primitive (advance, sidestep, squat, turn) and annotate the dominant cause. The failure patterns *align* with the locomotion tracking errors of the LMO controller reported in Table 4, where forward/lateral stepping exhibits larger position/orientation errors than turning and squatting, clarifying how its inaccuracies propagate to downstream manipulation failures.
>
> **Analysis**. The Sankey diagrams provide a stage-wise breakdown by first splitting failures into locomotion vs. pick/place errors and then into finer categories (e.g., object/basket unreachable, early stop, overshoot, wrong orientation, collisions, stumbles, grasp/placement errors). Across primitives, a large fraction of failures are locomotion-related—typically leaving the object or basket unreachable—while catastrophic events are rare, and squat shows a more balanced mix of locomotion and manipulation issues.

---

### Author Response · Authors · 2025-11-24

Dear AC,


We would like to flag a concern regarding reviewer **C1fR** that we strongly suspect that the review may have been generated by AI.

Please see the third-party consolidation report here: https://iclr.pangram.com/reviews?query=&submission_number=2007 (result: Fully AI-generated)

Besides, our observations are:
1. In **Suggestion 3**, the reviewer asks for “quantitative metrics (e.g., path/pose accuracy, time-to-goal, safety violations)”. For humanoid loco-manipulation setting, there is no **ground-truth path/pose**, so “path/pose accuracy” essentially **does not exist**. Moreover, *time-to-goal* and *safety violations* are almost never used in prior literature.
2. In **Suggestion 3**, the reviewer also requests “community-recognized scenarios”, which do **not** exist for real-world tasks.
3. In **Weakness 1** (“Insufficient Evidence for ‘Unified’ Whole-Body Control”), the reviewer criticizes a claim, we **never make**; the paper consistently talks about a “**Unified Latent VLA**” both in the title and in the text.
4. The weaknesses (“*Limited Generalization and Robustness Evaluation*”, “*No Failure Mode Analysis*”, “*Non-standardized Tasks & Metrics Hinder Comparability*”) are very generic, using common LLM-style tone without much practical, paper-specific analysis. And there is a lot of redundant and repetitive phrasing in the *Question* and *Suggestion*.

However, for the purpose of rebuttal and ICLR policy, we will still do our best to address concerns raised by this review. We respectfully remind AC / SAC / PC to assess the fairness of this review.


Best,

Authors

---

> ### Comment · Reviewer_C1fR · 2025-11-25
>
> I would like to clarify that the technical substance of my review was based on my own reading and analysis of the paper. I did use AI to help improve the clarity and wording of my written draft after forming my technical opinions, and this may have influenced the writing style. However, the concerns I raised reflected my own reasoning. Below I explain the points the authors highlighted.
>
> 1. Quantitative metrics.
>
> My comments about metrics were motivated by what I have seen in prior humanoid VLA work which is also cited by the authors. For example, [1] evaluates average positional error of joints, acceleration error, and velocity error. For whole-body loco-manipulation, I was also interested in time to reach target states and environment/self-collision behavior, as these reflect important aspects of safe, efficient and robust whole-body execution.
>
> 2. "Community-recognized scenarios."
>
> I acknowledge that my phrasing here was unclear. What I meant is that many whole-body humanoid tasks, especially in household or factory contexts, commonly include tasks like precise pick-and-place variations or door/drawer interactions. Some tasks in the paper, such as placing bags into a box or pushing a 50 kg cart, were harder for me to interpret in terms of their difficulty or broader significance.
>
> 3. "‘Unified’ Whole-Body Control"
>
> My concern was specifically about coordinated arm-leg motion, rather than the naming. In the paper’s demonstrations, most behaviors appear to involve sequential rather than simultaneous whole-body motion. For instance, pushing a cart mainly involves locomotion with the hands resting on the handle, which does not require complex arm–leg coordination. Likewise, in the bag pick-and-place task, the figures suggest that the robot performs picking while the legs remain still, and then walks with the arms held in a static state.
>
> 4. Generalization, robustness, failure modes, and comparability.
>
> It is true these categories are broad, but they are also common expectations for a high-quality paper submitted to ICLR. For a system intended to perform whole-body loco-manipulation in varied real-world environments, these evaluations help establish completeness and clarity.
>
> Overall, while AI was used to polish the writing, the review’s technical content was based entirely on my own understanding of the paper.
>
> [1]Pengxiang Ding, Jianfei Ma, Xinyang Tong, Binghong Zou, Xinxin Luo, Yiguo Fan, Ting Wang, Hongchao Lu, Panzhong Mo, Jinxin Liu, et al. Humanoid-VLA: Towards universal humanoid control with visual integration. arXiv preprint arXiv:2502.14795, 2025.

---

### Author Response · Authors · 2025-11-26
**General Author Response for Rebuttal**

Dear AC,

We appreciate your time and the additional effort required given the incident. We introduce WholeBodyVLA, a unified latent VLA system that enables large-space humanoid loco–manipulation in the real world by combining video-driven unified latent learning for VLA pretraining and a loco–manipulation–oriented RL controller for reliable lower-body execution.

Reviewers positively recognized our strengths and contributions, including the novelty of the unified latent learning design (*C1fR*, *N1iB*, *76f3*), the practicality of real-world performance (*C1fR*, *N1iB*, *76f3*, *1asV*), clear motivation and methodological structure (*N1iB*, *76f3*), diverse real-robot validation (*N1iB*, *76f3*, *1asV*), coherent integration of VLA with RL-based controller (*C1fR*, *N1iB*, *76f3*), and well-written (*C1fR*, *N1iB*).

Across all reviews, we note a common thread: the reviewers found the method conceptually sound and well-motivated, and the majority of concerns focused on **requests for additional experiments rather than technical issues.**

**Addressing Key Concerns.** We have made diligent efforts to directly address these requests and made corresponding modifications (highlighted in blue) in the updated manuscript:

- **Comprehensive Generalization Experiments:**  `C1fR(W2,Q2,S2), N1iB(W1,Q1,Q4), 76f3(W1), 1asV(W2,W3)`

A key concern is the generalization capability of our framework. We evaluated distance, object, and terrain generalization—as well as more challenging and diverse tasks such as long-horizon scenarios—demonstrating strong generalization and extensibility (Fig. 3, Sec. 4.3 & 4,5). See videos at `https://wholebodyvla.github.io/#generalization` [[**click here**](https://wholebodyvla.github.io/#generalization)] and below.


- **Scaling Benefits of Action-Free Video Pretraining:** `N1iB(W3,Q3), 1asV(W1,Q1)`

To clarify the contribution of action-free egocentric videos and their ability to reduce reliance on teleoperation data, we performed data-scaling experiments on latent learning, varying the amount of action-free video used for pretraining and the amount of teleoperation data used for finetuning. The results show that action-free videos effectively alleviate the scarcity of teleoperation data and substantially improve model performance (Fig. 3, Sec. 4.3). Moreover, our cross-domain latent visualizations (Fig. 8, Appendix C.5) provide examples of how this transfer occurs.

- **Clarity of Storyline and Explanations:** `1asV(discussion), N1iB(W2, Q2)`

We refined the writing to center the story on how egocentric videos are used to train WholeBodyVLA. We also add explanations of training protocol, baseline setups and refined the structure of experiments to ensure that the motivation, method, and experiments are tightly aligned.

- **Failure Case Analysis:** `C1fR(W4,Q1,S1)`

A related concern is the inherent difficulty of loco-manipulation and its common failure modes. Our analysis (Fig. 7, Appendix C.3) highlights key challenges—such as locomotion-induced instability, path deviation, and turn-while-advancing—and shows that these errors mainly arise from low-level execution rather than high-level decisions. This further motivates and validates the design of our LMO controller.


**Rebuttal Timeline and Status**
- Nov 24: We flagged concerns that the review of C1fR (score: 2) may be AI-generated due to factual and logical inconsistencies.
- Nov 26: We submitted detailed responses and new experimental results to all reviewers.
- Nov 27: Reviewer 1asV (score: 4) replied that he/she would "**conditional adjust score**" if we reformulated the paper story.
- Nov 28: The system closed before further reviewer follow-ups could be posted.
- Dec 3: We submitted a revised manuscript that updates the storyline in line with reviewer 1asV’s suggestion.

Thank you again for your time and careful consideration.

Best regards,
The Authors

---

### Meta-Review · Area_Chair_YfQe · 2026-01-04

**Summary:**

Reviewer C1fR has concerns such as (1) insufficient evidence for “unified” whole-body control,  (2) limited generalization and robustness evaluation, (3) non-standardized tasks & metrics hinder comparability, and (4) no failure mode analysis. These collectively reduce confidence in the paper’s significance and generality, leading the reviewer to recommend rejection.

Reviewer N1iB has concerns such as (1) ambiguity in loco-manipulation demands of the tasks, (2) baseline fairness, and (3) limited analysis of latent-learning/data scaling.

Reviewer 76f3 raises concerns that the demonstrated upper-body motions across the three tasks are relatively limited, and that the current tabletop task is comparatively simple and lacks diversity.

Reviewer 1asV 's concerns focus primarily on motivation clarity, methodological transparency, and evaluation breadth.

**Reviewer Concerns:**

The rebuttal has addressed major concerns, such as generalization capability, clarity of storyline and explanations, failure case analysis, and scaling benefits of action-free video pretraining, and lack of longer-horizon and more complex tasks.

The concern regarding non-standardized tasks and metrics limiting comparability is not fully addressed. However, this remains a long-standing challenge in the robotics community, as there are currently no widely accepted benchmarks for humanoid loco-manipulation. Prior related studies similarly rely on custom real-world setups. The Area Chair therefore does not view this limitation as sufficient grounds for rejection.

**Reviewer Scores:**

Reviewer C1fR would increase the rating to 4 (marginally below the acceptance threshold), as the rebuttal addressed only the major concerns.

Reviewer N1iB would increase their rating to 6 (marginally above the acceptance threshold), as the rebuttal satisfactorily resolved the raised issues.

Reviewer 76f3 would maintain the positive rating of 8, indicating continued support of the paper.

Reviewer 1asV would increase the rating to 6 (marginally above the acceptance threshold), as the rebuttal addressed the concerns and Reviewer 1asV has expressed the willingness to increase the rating after the rebuttal.

Overall, the rebuttal has substantially addressed the major concerns raised by the reviewers. Based on the updated paper and the rebuttal, the AC recommends acceptance of the revised paper.

---

### Decision · Program_Chairs · 2026-01-26

Accept (Poster)